# Prosapip1 (encoded by the *Lzts3* gene) in the dorsal hippocampus mediates synaptic protein composition, long-term potentiation, and spatial memory

Zachary W Hoisington[1], Himanshu Gangal[2], Khanhky Phamluong[1], Chhavi Shukla[1], Jeffrey J Moffat[1], Alexandra Salvi[1], Gregg Homanics[3], Jun Wang[2], Yann Ehinger[1]*, Dorit Ron[1]*

[1]Alcohol and Addiction Research Group, Department of Neurology, University of California, San Francisco, San Francisco, United States; [2]Department of Neuroscience and Experimental Therapeutics, School of Medicine, Texas A&M University Health Science Center, College Station, United States; [3]Department of Anesthesiology and Perioperative Medicine, University of Pittsburgh, Pittsburgh, United States

*For correspondence:
yann.ehinger@gmail.com (YE);
dorit.ron@ucsf.edu (DR)

Competing interest: The authors declare that no competing interests exist.

## eLife Assessment

This **important** study aims to understand the function of ProSAP-interacting protein 1 (Prosapip1) in the brain. Using a conditional Prosapip1 KO mouse (floxed prosapip1 crossed with Syn1-Cre line), the authors performed analysis including protein biochemistry, synaptic physiology, and behavioral learning. **Convincing** evidence from this study supports a role of Prosapip 1 in synaptic protein composition, synaptic NMDA responses, LTP, and spatial memory.

**Abstract**    Prosapip1 is a brain-specific protein, encoded by *Lzts3*, localized to the postsynaptic density, where it promotes dendritic spine maturation in primary hippocampal neurons. However, nothing is known about the role of Prosapip1 in vivo. To examine this, we utilized the Cre-loxP system to develop a Prosapip1 neuronal knockout mouse. We found that Prosapip1 controls the synaptic localization of its binding partner SPAR, along with PSD-95 and the GluN2B subunit of the NMDA receptor (NMDAR) in the dorsal hippocampus (dHP). We next sought to identify the potential contribution of Prosapip1 to the activity and function of the NMDAR and found that Prosapip1 plays an important role in NMDAR-mediated transmission and long-term potentiation (LTP) in the CA1 region of the dHP. As LTP is the cellular hallmark of learning and memory, we examined the consequences of neuronal knockout of Prosapip1 on dHP-dependent memory. We found that global or dHP-specific neuronal knockout of Prosapip1 caused a deficit in learning and memor,y whereas developmental, locomotor, and anxiety phenotypes were normal. Taken together, Prosapip1 in the dHP promotes the proper localization of synaptic proteins which, in turn, facilitates LTP driving recognition, social, and spatial learning and memory.

## Introduction

The brain-specific protein ProSAP-interacting protein 1 (Prosapip1), encoded by the gene *Lzts3,* shows particularly high expression in the cortex, cerebellum, and hippocampus (*Wendholt et al., 2006*; *Laguesse et al., 2017*). Prosapip1 is highly enriched in the PSD of excitatory synapses and plays an

important role in dendritic spine maturation in primary hippocampal neurons (**Wendholt et al., 2006**; **Dolnik et al., 2016**). The PSD is a complex protein network located at the postsynaptic membrane of neurons, namely excitatory neurons (**Boeckers et al., 2002**; **Kaizuka and Takumi, 2018**). The PSD is crucial for synaptic transmission and plasticity (**Boeckers et al., 2002**; **Kaizuka and Takumi, 2018**), and abnormalities in the PSD network are linked to various neuropsychiatric and neurodegenerative disorders (**Kaizuka and Takumi, 2018**), such as schizophrenia, autism spectrum disorder (ASD), depression, and Alzheimer's disease (**Kaizuka and Takumi, 2018**). The PSD contains scaffolding proteins such as PSD-95, Shank, and Homer (**Kennedy, 2018**). The major scaffold in the PSD is actin (**Qualmann et al., 2004**), which recycles between globular (G-actin) and filamentous (F-actin) actin and is crucial in reorganizing and stabilizing the PSD (**Qualmann et al., 2004**; **Kuriu et al., 2006**).

Prosapip1 has a C-terminal Fez1 domain that it shares with the other members of the Fezzin family (LAPSER1, PSD-Zip70, N4BP3) (**Wendholt et al., 2006**). Fezzins interact with the PDZ (PSD-95/Dlg1/ZO-1) domain of SH3 and multiple ankyrin repeat domains (Shank) family proteins. Shanks are also known as proline-rich synapse-associated proteins (ProSAPs) and were originally identified as proteins primarily localizing in the PSD of excitatory synapses (**Boeckers et al., 1999**; **Sheng and Kim, 2000**; **Boeckers et al., 2002**). Prosapip1 specifically interacts with the PSD protein, Shank3 (**Wendholt et al., 2006**). Prosapip1 also interacts with the spine-associated Rap GTPase-activating protein (SPAR) (**Wendholt et al., 2006**; **Reim et al., 2016**). SPAR catalyzes the conversion of the small G-protein Rap from its active, GTP-bound form to its inactive GDP-bound form, thereby inhibiting Rap activity. SPAR is an essential modulator of spine morphology through its interaction with actin (**Pak et al., 2001**; **Richter et al., 2007**). In hippocampal neurons, it is localized to dendritic spines and leads to spine head enlargement (**Pak et al., 2001**). SPAR also associates with PSD-95/NMDAR-complex components (**Matsuura et al., 2022**). Wendholt et al. reported that in primary hippocampal neurons Prosapip1 regulates the synaptic levels of SPAR by scaffolding it to Shank3 (**Wendholt et al., 2006**; **Dolnik et al., 2016**; **Reim et al., 2016**).

We previously found that mRNA to protein translation of Prosapip1 is controlled by the mechanistic target of rapamycin complex 1 (mTORC1) in the nucleus accumbens (NAc) of mice consuming large quantities of alcohol (**Laguesse et al., 2017**). We further showed that Prosapip1 interacts with SPAR in the NAc and leads to alcohol-dependent synaptic and structural plasticity (**Laguesse et al., 2017**). Specifically, we found that knockdown of Prosapip1 in the NAc led to a reduction in F-actin, while overexpression of Prosapip1 led to an increase in F-actin (**Laguesse et al., 2017**). Knockdown of Prosapip1 caused a reduction of mature, mushroom-type spines and an increase in immature, thin spines (**Laguesse et al., 2017**). Finally, we found that Prosapip1 in the NAc contributes to alcohol self-administration and reward (**Laguesse et al., 2017**). However, nothing is known about the normal in vivo role of Prosapip1 in the central nervous system (CNS). Here, we report that Prosapip1 in the dHP controls the localization of synaptic proteins at the PSD and contributes to LTP as well as recognition, social, and spatial learning and memory.

## Results

### Generation and characterization of *Prosapip1*(fl/fl) mice

To elucidate the role of Prosapip1 in the CNS, we generated a mouse line with flox sites flanking the Prosapip1 encoding gene, *Lzts3*. Guide RNA binding sites were identified in intron 2 and in the 3' UTR of exon 5 of *Lzts3* (**Figure 1A**). PAGE-purified Ultramer single-stranded DNA oligos that were homologous to the target loci in intron 2 and exon 5 (**Figure 1B**) were used as repair templates.

To generate a neuron-specific conditional knockout of Prosapip1, we crossed homozygous *Lzts3* floxed (*Lzts3*fl/fl) mice with a Synapsin I promoter–driven Cre recombinase line (Syn1-Cre). Specifically, male *Lzts3*fl/fl;Syn1-Cre(–) mice were bred with female *Lzts3*fl/fl;Syn1-Cre(+) mice (**Figure 1C**), producing *Lzts3*fl/fl;Syn1-Cre(+) offspring (Prosapip1 cKO) and *Lzts3*fl/fl;Syn1-Cre(–) littermates, which served as controls. The neuronal knockout was confirmed by estern blot analysis using an anti-Prosapip1 antibody. As shown in **Figure 1D**, control mice have Prosapip1 protein levels comparable to those of a C57BL/6 mouse, while Prosapip1 cKO mice show a complete knockout of Prosapip1 protein in the dHP.

We examined the breeding history of the newly developed line to identify any abnormalities. The litter sizes of *Lzts3*fl/fl;Syn1-Cre matings followed a normal distribution with an average litter size

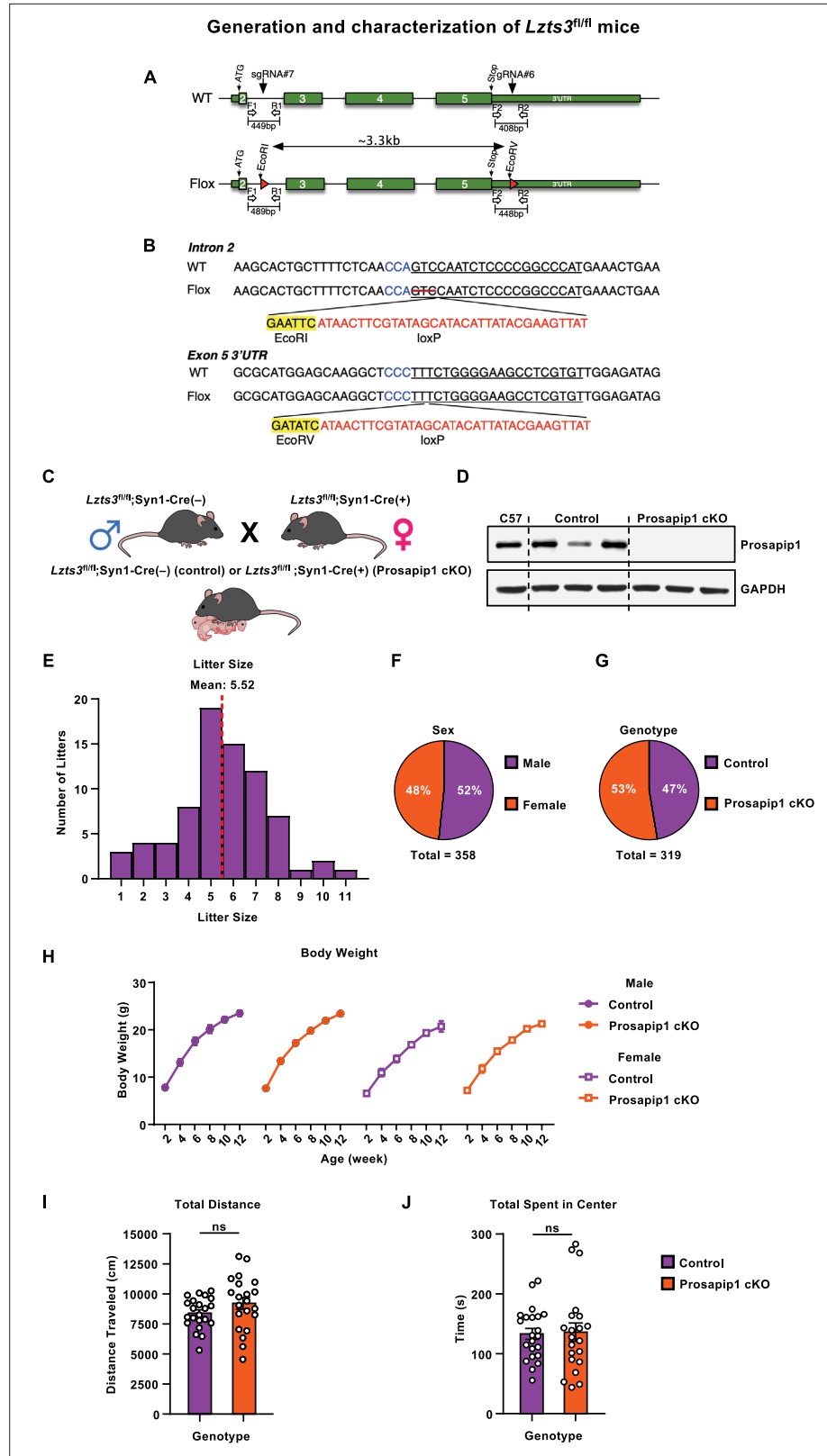

**Figure 1.** Generation and characterization of Prosapip1 cKO mice. (**A**) Identification of guide RNA binding sites in intron 2 and the 3' UTR of exon 5 of *Lzts3*. (**B**) PAGE-purified Ultramer single-stranded DNA oligos that were homologous to the target loci in intron 2 and exon 5 were used as repair templates. (**C**) Genetic crossing scheme of the *Lzts3*fl/fl;Syn1-Cre mice. Male *Lzts3*fl/fl;Syn1-Cre(-) mice were mated with female *Lzts3*fl/fl;Syn1-Cre(+) (Prosapip1

*Figure 1 continued on next page*

*Figure 1 continued*

cKO) mice, leading to litters of control (*Lzts3*fl/fl;Syn1-Cre(-)) or Prosapip1 cKO (*Lzts3*fl/fl;Syn1-Cre(+)) mice. (**D**) The dHP of C57BL/6, control and Prosapip1 cKO mice was dissected. Prosapip1 was detected using anti-Prosapip1 antibodies. GAPDH was used as a loading control. (**E**) Histogram of litter size from *Lzts3*fl/fl × Syn1-Cre mating pairs. X-axis depicts number of pups per litter, while Y-axis depicts numbers of litters at that size. (**F**) Proportion of male and female offspring from *Lzts3*fl/fl × Syn1-Cre matings. (**G**) Proportion of Cre(-) and Cre(+) offspring from *Lzts3*fl/fl × Syn1-Cre matings. (**H**) Body weight of male and female Prosapip1 cKO and control mice was measured biweekly from birth to assess overt developmental deficits. Data are represented as mean ± SEM and analyzed using three-way ANOVA (*Table 1*). n=10 (control male), 5 (control female), 11 (Prosapip1 cKO male), 11 (Prosapip1 cKO female). (**I–J**) Mice were placed in an open field and locomotion was recorded for 20 min. Total distance traveled during the open field test (**I**) and time spent in the center of the open field (**J**). Data represented as mean ± SEM and analyzed using an unpaired t-test (*Table 1*). ns, non-significant. n=22 (control), 21 (Prosapip1 cKO).

The online version of this article includes the following source data for figure 1:

**Source data 1.** File containing raw data for *Figure 1*.

**Source data 2.** Original files for western blot analysis displayed in *Figure 1D*.

between 5 and 6 pups (*Figure 1E*, *Table 1*). The proportion of male and female pups were near expectations, with 52% of pups being male and 48% female (*Figure 1F*). Inheritance of Syn1-Cre(+) was also split evenly, with 53% of pups from the line being born with the transgene (*Figure 1G*). Body weight of male and female mice was monitored every 2 weeks from birth to assess potential developmental effects. While males were significantly heavier than females throughout development, there was no significant effect of the genotype on weight over time (*Figure 1H*, *Table 1*).

We next assessed baseline locomotor and exploratory behavior of these mice. Upon reaching adulthood, mice were placed in an open field and allowed to explore for 20 min, and total distance traveled and time spent in the center of the open field were measured. Deletion of neuronal Prosapip1 did not alter locomotor activity (*Figure 1I*, *Table 1*). Furthermore, there were no differences between genotypes when assessing the time spent in the center of the open field (*Figure 1J*, *Table 1*).

## Prosapip1 is required for synaptic localization of PSD proteins

Prosapip1 is a scaffolding protein that recruits SPAR to the PSD and interconnects it with Shank3 in primary hippocampal neurons (*Wendholt et al., 2006*; *Reim et al., 2016*). First, we examined if the synaptic localization of SPAR was affected by Prosapip1 neuronal knockout. To do this, we isolated the synaptic fraction from the total homogenate. We confirmed the efficacy of the crude synaptosomal fraction isolation by measuring enrichment of synapsin, a protein found at the synapse, and lack of cAMP response element-binding protein (CREB), a transcription factor located in the nucleus (*Figure 2A*). We analyzed the total (*Figure 2B*) and synaptic (*Figure 2C*) levels of SPAR in the dHP of control and Prosapip1 cKO male and female mice. Total levels of SPAR were similar between the groups (*Figure 2B*, *Table 1*). However, we detected a significant reduction in synaptic SPAR in Prosapip1 cKO mice as compared to the control group (*Figure 2C*, *Table 1*).

Prosapip1 interconnects SPAR to Shank3, and SPAR forms a complex with PSD-95 (*Wendholt et al., 2006*; *Reim et al., 2016*; *Matsuura et al., 2022*). Therefore, we next examined the total and synaptic levels of Shank3 and PSD-95. Prosapip1 deletion did not affect the total or synaptic levels of Shank3 (*Figure 2D–G*, *Table 1*). We also found that Prosapip1 cKO mice have similar levels of total PSD-95 (*Figure 2D and H*, *Table 1*) compared to controls. However, when examining synaptic PSD-95, Prosapip1 cKO mice displayed drastically reduced levels compared to controls (*Figure 2E and I*, *Table 1*). Collectively, these results imply that Prosapip1 in the dHP is necessary for synaptic localization of SPAR and PSD-95, but not Shank3. It is important to note that our method of synaptic fractionation does not distinguish between pre- and post-synaptic compartments. Consequently, we cannot determine whether Prosapip1 is involved in the assembly of axonal proteins.

SPAR and Shank3 are both known to interact with the NMDAR and AMPAR, and PSD-95 stabilizes NMDAR surface localization (*Naisbitt et al., 1999*; *Tu et al., 1999*; *Pak et al., 2001*; *Boeckers et al., 2002*; *Arons et al., 2012*; *Won et al., 2016*; *Coley and Gao, 2019*; *Matsuura et al., 2022*). As the synaptic localization of SPAR and PSD-95 is disrupted in the absence of Prosapip1, we next tested the synaptic levels of NMDAR subunits GluN2A and GluN2B, along with AMPAR subunit GluA1. There was no change in the overall levels of NMDAR subunits GluN2A and GluN2B or the AMPAR subunit

**Table 1.** Statistics.

| Figure | n | Sex difference | Statistical test | Effect | Result | p-value | Post-hoc | Post-hoc comparison | p-value |
|--------|---|----------------|------------------|--------|--------|---------|----------|---------------------|---------|
| *Figure 1E* | 77 | | D'Agostino & Pearson test | Normality | K2=5.480 | 0.0646 | | | |
| *Figure 1H* | 10 (Cre- Male), 5 (Cre-Female), 11 (Cre +Male), 11 (Cre +Female) | Yes | Three-way ANOVA | Genotype | $F_{(1,33)}=0.7842$ | 0.3823 | | | |
| | | | | Sex | $F_{(1,33)}=23.94$ | <0.0001 | | | |
| | | | | Time | $F_{(5,158)}=588.4$ | <0.0001 | | | |
| | | | | Genotype x Sex | $F_{(1,33)}=1.345$ | 0.2545 | | | |
| | | | | Genotype x Time | $F_{(5,158)}=0.1476$ | 0.9805 | | | |
| | | | | Sex x Time | $F_{(5,158)}=2.345$ | 0.0437 | | | |
| | | | | Genotype x Sex x Time | $F_{(5,158)}=0.3764$ | 0.8643 | | | |
| *Figure 1I* | 22 (Cre-), 21 (Cre+) | No | Unpaired t-test | Genotype | $t_{(41)}=1.460$ | 0.1518 | | | |
| *Figure 1J* | 22 (Cre-), 21 (Cre+) | No | Mann-Whitney Test | Genotype | U=220 | 0.8007 | | | |
| *Figure 2B* | 5 (Cre-), 5 (Cre+) | | Unpaired t-test | Genotype | $t_{(6.963)}=1.427$ | 0.1968 | | | |
| *Figure 2C* | 5 (Cre-), 5 (Cre+) | | Unpaired t-test | Genotype | $t_{(5.880)}=5.183$ | 0.0022 | | | |
| *Figure 2F* | 5 (Cre-), 5 (Cre+) | | Unpaired t-test | Genotype | $t_{(6.932)}=1.345$ | 0.221 | | | |
| *Figure 2G* | 5 (Cre-), 5 (Cre+) | | Unpaired t-test | Genotype | $t_{(6.720)}=0.1306$ | 0.8999 | | | |
| *Figure 2H* | 9 (Cre-), 10 (Cre+) | | Unpaired t-test | Genotype | $t_{(16.38)}=0.5588$ | 0.5839 | | | |
| *Figure 2I* | 9 (Cre-), 10 (Cre+) | | Unpaired t-test | Genotype | $t_{(16.44)}=6.095$ | <0.0001 | | | |
| *Figure 2L* | 4 (Cre-), 5 (Cre+) | | Unpaired t-test | Genotype | $t_{(4.429)}=0.1050$ | 0.921 | | | |
| *Figure 2M* | 4 (Cre-), 5 (Cre+) | | Unpaired t-test | Genotype | $t_{(6.689)}=0.1758$ | 0.8656 | | | |
| *Figure 2N* | 4 (Cre-), 5 (Cre+) | | Unpaired t-test | Genotype | $t_{(6.700)}=1.675$ | 0.1398 | | | |
| *Figure 2O* | 4 (Cre-), 5 (Cre+) | | Unpaired t-test | Genotype | $t_{(6.332)}=5.524$ | 0.0012 | | | |
| *Figure 2P* | 4 (Cre-), 5 (Cre+) | | Unpaired t-test | Genotype | $t_{(6.956)}=0.1831$ | 0.8599 | | | |
| *Figure 2Q* | 4 (Cre-), 5 (Cre+) | | Unpaired t-test | Genotype | $t_{(6.060)}=0.8548$ | 0.4252 | | | |
| *Figure 3D* | 10/6 (Cre-), 9/5 (Cre+) | | Unpaired t-test | Genotype | $t_{(17)}=3.933$ | 0.0011 | | | |

*Table 1 continued on next page*

*Table 1 continued*

| Figure | n | Sex difference | Statistical test | Effect | Result | p-value | Post-hoc | Post-hoc comparison | p-value |
|---|---|---|---|---|---|---|---|---|---|
| *Figure 3E* | 14/4 (Cre-), 11/4 (Cre+) | | Two-way ANOVA | Genotype | $F_{(1,3)}$=9.441 | 0.044 | Tukey | 0.2 (Cre- vs. Cre+) | 0.345 |
| | | | | Intensity | $F_{(1,3)}$=9.441 | <0.001 | | 0.4 (Cre- vs. Cre+) | 0.025 |
| | | | | Genotype x Intensity | $F_{(1,3)}$=1.376 | 0.258 | | 0.6 (Cre- vs. Cre+) | 0.034 |
| | | | | | | | | 0.9 (Cre- vs. Cre+) | 0.046 |
| *Figure 3F* | 3/4 (Cre-), 11/4 (Cre+) | | Unpaired t-test | Genotype | $t_{(22)}$=3.094 | 0.00529 | | | |
| *Figure 3G* | 11/3 (Cre-), 13/4 (Cre+) | | Unpaired t-test | Genotype | $t_{(22)}$=2.099 | 0.0475 | | | |
| *Figure 4A* | 22 (Cre-), 20 (Cre+) | No | Two-way ANOVA | Genotype | $F_{(1,40)}$=0.3079 | $P$=0.5821 | Šidák | Familiar vs. Novel (Cre-) | <0.0001 |
| | | | | Object | $F_{(1,40)}$=34.32 | <0.0001 | | Familiar vs. Novel (Cre+) | 0.4871 |
| | | | | Genotype x Object | $F_{(1,40)}$=18.37 | 0.0001 | | | |
| *Figure 4B* | 22 (Cre-), 19 (Cre+) | No | Unpaired t-test | Genotype | $t_{(26.08)}$=5.434 | <0.0001 | | | |
| *Figure 4D* | 14 (Cre-), 17 (Cre+) | No | Two-way ANOVA | Genotype | $F_{(1,29)}$=5.419 | 0.0271 | Šidák | Empty vs. Social (Cre-) | 0.0006 |
| | | | | Social | $F_{(1,29)}$=34.30 | <0.0001 | | Empty vs. Social (Cre+) | 0.0005 |
| | | | | Genotype x Social | $F_{(1,29)}$=0.05655 | 0.8137 | | | |
| *Figure 4E* | 14 (Cre-), 17 (Cre+) | No | Two-way ANOVA | Genotype | $F_{(1,29)}$=1.463 | 0.2362 | Šidák | Familiar vs. Novel (Cre-) | 0.0008 |
| | | | | Social Novelty | $F_{(1,29)}$=17.60 | 0.0002 | | Familiar vs. Novel (Cre+) | 0.1451 |
| | | | | Genotype x Social Novelty | $F_{(1,29)}$=2.947 | 0.0967 | | | |
| Ext. *Figure 4A* | 6 (Cre-), 12 (Cre+) | No | Unpaired t-test | Genotype | $t_{(16)}$=0.2635 | 0.7955 | | | |

*Table 1 continued on next page*

*Table 1 continued*

| Figure | n | Sex difference | Statistical test | Effect | Result | p-value | Post-hoc | Post-hoc comparison | p-value |
|---|---|---|---|---|---|---|---|---|---|
| *Figure 4F* | 9 (Cre-), 12 (Cre+) | No | Two-way ANOVA | Genotype | $F_{(1,19)}=102.0$ | <0.0001 | Šidák | Trial 2 (Cre- vs. Cre+) | <0.0001 |
| | | | | Trial | $F_{(15,285)}=7.899$ | <0.0001 | | Trial 3 (Cre- vs. Cre+) | 0.0067 |
| | | | | Genotype x Trial | $F_{(15,285)}=1.401$ | 0.1454 | | Trial 4 (Cre- vs. Cre+) | 0.0018 |
| | | | | | | | | Trial 5 (Cre- vs. Cre+) | 0.0002 |
| | | | | | | | | Trial 6 (Cre- vs. Cre+) | 0.002 |
| | | | | | | | | Trial 7 (Cre- vs. Cre+) | 0.0002 |
| | | | | | | | | Trial 8 (Cre- vs. Cre+) | 0.0016 |
| | | | | | | | | Trial 9 (Cre- vs. Cre+) | <0.0001 |
| | | | | | | | | Trial 10 (Cre- vs. Cre+) | 0.0006 |
| | | | | | | | | Trial 11 (Cre- vs. Cre+) | 0.0049 |
| | | | | | | | | Trial 12 (Cre- vs. Cre+) | 0.0169 |
| | | | | | | | | Trial 13 (Cre- vs. Cre+) | 0.0098 |
| *Figure 4G* | 9 (Cre-), 12 (Cre+) | No | Three-way ANOVA | Genotype | $F_{(1,19)}=100.7$ | <0.0001 | | | |
| | | | | Trial | $F_{(15,289)}=7.289$ | <0.0001 | | | |
| | | | | Error Type | $F_{(0.4468, 8.489)}=4.554$ | 0.0772 | | | |
| | | | | Genotype x Trial | $F_{(15,289)}=1.529$ | 0.094 | | | |
| | | | | Genotype x Error Type | $F_{(1,19)}=15.09$ | 0.001 | | | |
| | | | | Trial x Error Type | $F_{(6.157, 117)}=2.865$ | 0.0115 | | | |
| | | | | Genotype x Trial x Error Type | $F_{(15,285)}=1.831$ | 0.0303 | | | |
| *Figure 4K* | 9 (Cre-), 12 (Cre+) | No | Mann-Whitney Test | Genotype | U=8 | 0.0004 | | | |
| Ext. *Figure 4B* | 15 (Cre-), 16 (Cre+) | No | Two-way ANOVA | Chamber | $F_{(1,29)}=92.69$ | <0.0001 | Šidák | Light vs. Dark (Cre-) | <0.0001 |
| | | | | Genotype | $F_{(1,29)}=0.9504$ | 0.3377 | | Light vs. Dark (Cre+) | <0.0001 |
| | | | | Genotype x Chamber | $F_{(1,29)}=0.09645$ | 0.7584 | | | |
| Ext. *Figure 4C* | 7 (Cre-), 17 (Cre+) | No | Mann-Whitney Test | Genotype | U=42 | 0.2878 | | | |
| *Figure 5C* | 19 (AAV-GFP), 15 (AAV-Cre) | No | Unpaired t-test | Treatment | $t_{(29.59)}=1.961$ | 0.0594 | | | |

*Table 1 continued on next page*

*Table 1 continued*

| Figure | n | Sex difference | Statistical test | Effect | Result | p-value | Post-hoc | Post-hoc comparison | p-value |
|---|---|---|---|---|---|---|---|---|---|
| *Figure 5D* | 19 (AAV-GFP), 15 (AAV-Cre) | No | Unpaired t-test | Treatment | $t_{(29.72)}=1.066$ | 0.2951 | | | |
| *Figure 5E* | 19 (AAV-GFP), 15 (AAV-Cre) | No | Two-way ANOVA | Treatment | $F_{(1,32)}=2.361$ | 0.1342 | Šidák | Light vs. Dark (AAV-GFP) | <0.0001 |
| | | | | Chamber | $F_{(1,32)}=46.74$ | <0.0001 | | Light vs. Dark (AAV-Cre) | 0.0003 |
| | | | | Treatment x Chamber | $F_{(1,32)}=0.1588$ | 0.6929 | | | |
| *Figure 5F* | 16 (AAV-GFP), 17 (AAV-Cre) | No | Two-way ANOVA | Treatment | $F_{(1,31)}=3.069$ | 0.0897 | Šidák | Familiar vs. Novel (AAV-GFP) | <0.0001 |
| | | | | Object | $F_{(1,31)}=36.30$ | <0.0001 | | Familiar vs. Novel (AAV-Cre) | 0.9965 |
| | | | | Treatment x Object | $F_{(1,31)}=35.05$ | <0.0001 | | | |
| *Figure 5G* | 18 (AAV-GFP), 15 (AAV-Cre) | No | Unpaired t-test | Treatment | $t_{(26.04)}=3.777$ | 0.0008 | | | |
| *Figure 5H* | 18 (AAV-GFP), 15 (AAV-Cre) | No | Two-way ANOVA | Treatment | $F_{(1,31)}=0.0003957$ | 0.9843 | Šidák | Empty vs. Social (AAV-GFP) | <0.0001 |
| | | | | Social | $F_{(1,31)}=31.60$ | <0.0001 | | Empty vs. Social (AAV-Cre) | 0.0062 |
| | | | | Treatment x Social | $F_{(1,31)}=0.7781$ | 0.3845 | | | |
| *Figure 5I* | 18 (AAV-GFP), 15 (AAV-Cre) | No | Two-way ANOVA | Treatment | $F_{(1,31)}=0.2045$ | 0.6543 | Šidák | Familiar vs. Novel (AAV-GFP) | 0.0303 |
| | | | | Social Novelty | $F_{(1,31)}=9.777$ | 0.0038 | | Familiar vs. Novel (AAV-Cre) | 0.1319 |
| | | | | Treatment x Social Novelty | $F_{(1,31)}=0.1131$ | 0.7389 | | | |
| *Figure 5J* | 8 (AAV-GFP), 6 (AAV-Cre) | No | Two-way ANOVA | Treatment | $F_{(1,12)}=22.95$ | 0.0004 | Šidák | Trial 5 (AAV-GFP vs. AAV-Cre) | <0.0001 |
| | | | | Trial | $F_{(15,180)}=6.246$ | <0.0001 | | Trial 9 (AAV-GFP vs. AAV-Cre) | 0.0026 |
| | | | | Treatment x Trial | $F_{(15,180)}=2.033$ | 0.0153 | | Trial 10 (AAV-GFP vs. AAV-Cre) | <0.0001 |
| | | | | | | | | Trial 14 (AAV-GFP vs. AAV-Cre) | 0.0475 |
| *Figure 5K* | 8 (AAV-GFP), 6 (AAV-Cre) | No | Mann-Whitney Test | Treatment | U=3 | 0.0047 | | | |

GluA1 (*Figure 2J, L, N and P*, *Table 1*). The synaptic levels of GluN2A and GluA1 were also unaltered in the dHP of Prosapip1 cKO mice compared to controls (*Figure 2K, M and Q*, *Table 1*). In contrast, the synaptic levels of the NMDAR subunit GluN2B were significantly reduced in Prosapip1 cKO mice (*Figure 2K and Q*, *Table 1*). Together, these data suggest that Prosapip1 controls the synaptic localization of NMDAR subunit GluN2B, but not GluN2A or GluA1, in the dHP.

Further studies are needed to determine the mechanism by which Prosapip1 controls the localization of PSD95,GluN2BB, and potentially others. It is plausible that post-translational modifications are responsible for Prosapip1 function. For example, the Prosapip1 sequence contains a potential glycosylation site (Ser622) and several potential phosphorylation sites. These post-translational modifications can contribute to the stabilization of the synaptic localization of GluN2B and PSD95. Therefore, proteomic studies are required for a comprehensive study of the role of Prosapip1 in the PSD.

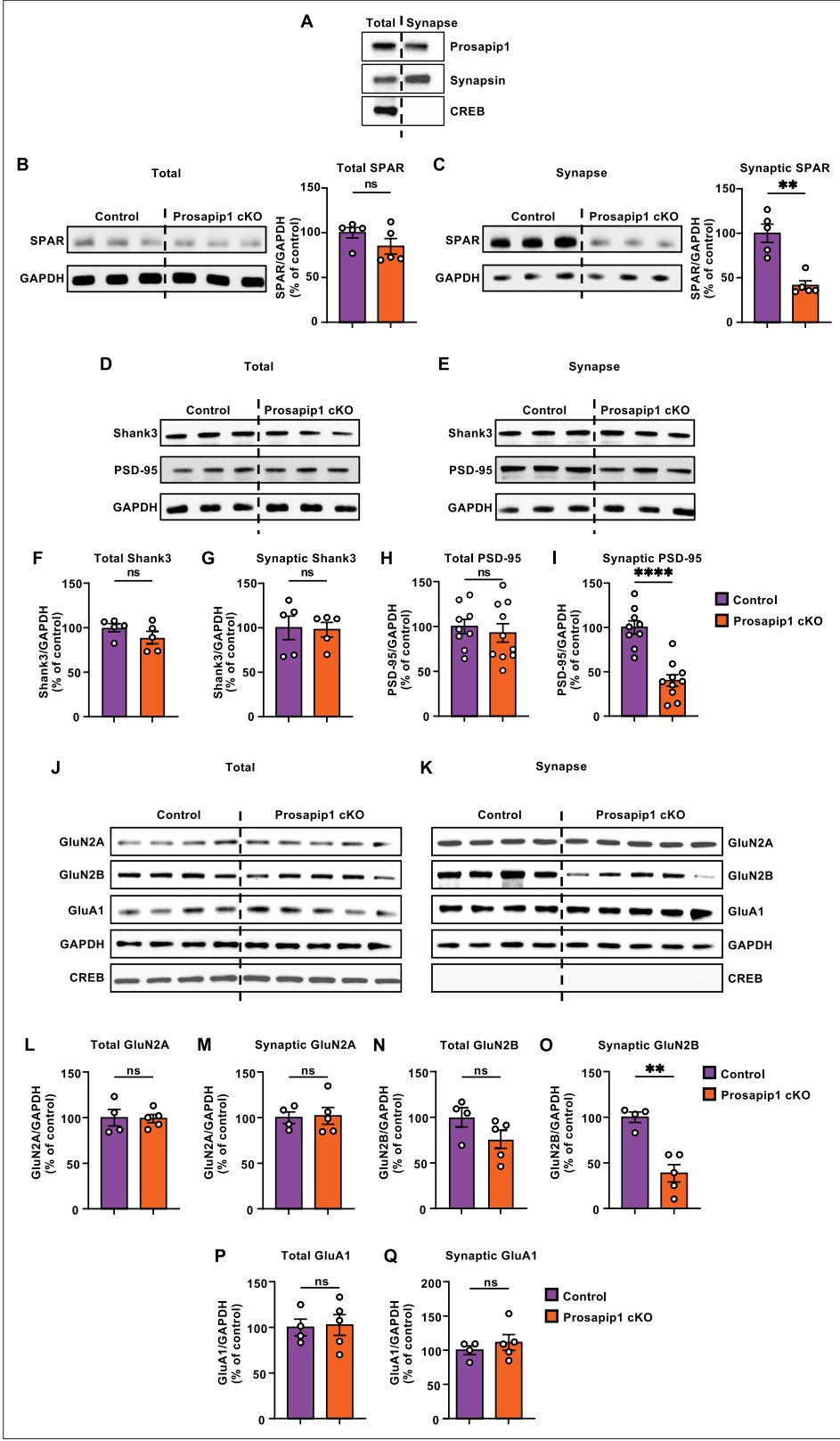

**Figure 2.** Prosapip1 is required for synaptic localization of PSD proteins. (**A**) The levels of Prosapip1 (top), Synapsin (middle), and cAMP response element-binding protein (CREB) (bottom) in the dorsal hippocampus of control (*Lzts3*^fl/fl^;Syn1-Cre(-)) mice were measured in the total and crude synaptosomal fraction. (**B–I**) Total levels of SPAR (**B**), Shank3 (**D, F**), and PSD-95 (**D, H**) alongside the synaptic levels of SPAR (**C**), Shank3 (**E, G**), and PSD-95

*Figure 2 continued on next page*

*Figure 2 continued*

(**E, I**) were measured in the dHP of Prosapip1 cKO and control mice using western blot analysis. Protein levels were normalized to GAPDH and presented as a percentage of the average of the control mice values. Data are represented as mean ± SEM and analyzed using an unpaired two-tailed t-test with Welch's correction (*Table 1*). **p<0.01, ****p<0.0001; ns, non-significant. n=5 per group (**B–G**), 9 (control) and 10 (Prosapip1 cKO) (**H–I**). (**J–Q**) The total levels of GluN2A (**J, L**), GluN2B (**J, N**), and GluA1 (**J, P**) and the synaptic levels of GluN2A (**K, M**), GluN2B (**K, O**), and GluA1 (**K, Q**) were measured in the dHP of Prosapip1 cKO and control mice using western blot analysis. Protein levels were normalized to GAPDH and presented as a percentage of the average of the control mice values. Data are represented as mean ± SEM and analyzed using an unpaired two-tailed t-test with Welch's correction (*Table 1*). **p<0.01; ns, non-significant. n=4 (control), 5 (Prosapip1 cKO).

The online version of this article includes the following source data and figure supplement(s) for figure 2:

**Source data 1.** File containing raw data for *Figure 2*.

**Source data 2.** Original files for western blot analysis displayed in *Figure 2A–E, J and K*.

**Figure supplement 1.** Verification of the crude synaptic preparation.

**Figure supplement 1—source data 1.** File containing raw data for *Figure 2—figure supplement 1*.

**Figure supplement 1—source data 2.** Original files for western blot analysis displayed in *Figure 2—figure supplement 1B*.

## Prosapip1 in the dorsal hippocampus plays a role in NMDA receptor-mediated transmission and long-term potentiation

The GluN2B subunit of the NMDAR is critically involved in learning and memory, a process reliant on LTP (*Nachtigall et al., 2024*). Therefore, we examined the contribution of Prosapip1 to GluN2B-mediated LTP. To examine this, hippocampal slices from male and female Prosapip1 cKO and control mice were prepared for electrophysiology recordings. Bipolar stimulating electrodes were positioned in the CA1 region of the dHP. Simultaneously, the recording electrode was placed approximately 100–150 µm away from the stimulating electrode (*Figure 3A*). The administration of a single electrical pulse (2ms) elicited a fiber volley, followed by a field excitatory postsynaptic potential response (fEPSPs) (*Figure 3B*). Before inducing LTP, a stable baseline of fEPSPs was established for 10 min (*Figure 3C*). Upon establishing a baseline of fEPSPs, high-frequency stimulation (HFS) was applied at 100 Hz for 1 s, repeated four times at 20 s intervals, to trigger LTP. In the control mice, there was a significant increase in the fEPSP amplitude following HFS, indicating a successful LTP induction (*Figure 3C*), which persisted for at least 30 min (*Figure 3C*). However, LTP was absent in the Prosapip1 cKO mice (*Figure 3C*). A comparative analysis of the normalized fEPSPs, conducted 20–30 min post-HFS, revealed a significantly reduced fEPSP amplitude in the Prosapip1 cKO group compared to controls (*Figure 3D*, *Table 1*). Together, these findings strongly suggest Prosapip1 plays an important role in dHP LTP.

LTP induction in the hippocampus is known to rely on the functioning of postsynaptic NMDARs. Therefore, we aimed to discern if there were any differences between Prosapip1 cKO and control mice in terms of NMDAR-mediated excitatory postsynaptic currents (EPSCs) in dHP CA1 pyramidal neurons. A whole-cell voltage-clamp recording technique was employed for this purpose. First, a low concentration of external $Mg^{2+}$ (0.05 mM) was used to facilitate the removal of the magnesium block from the NMDAR channel. Concurrently, AMPARs and $GABA_A$ receptors ($GABA_A$Rs) were pharmacologically inhibited. NMDAR-EPSCs were recorded in response to an escalating series of stimulus intensities. The amplitude of NMDAR-EPSCs was significantly reduced in the Prosapip1 cKO mice compared to the controls (*Figure 3E*, *Table 1*). Next, we investigated whether LTP inhibition because of Prosapip1 deletion is due to post and/or presynaptic changes. First, we investigated postsynaptic changes by measuring NMDA-induced currents. NMDA (20 µM) was bath applied for 30 s and peak current was measured. We observed that the peak current in the Prosapip1 cKO mice was markedly lower than that in their control counterparts (*Figure 3F*, *Table 1*). These data suggest that postsynaptic NMDAR responsiveness was decreased in the dHP of Prosapip1 cKO mice as compared to control mice. Finally, to examine the presence of any presynaptic alterations in glutamatergic transmission in CA1 neurons upon Prosapip1 deletion, we measured the paired-pulse ratio (PPR, 100 millisecond interval) of electrically evoked EPSCs in CA1 dHP neurons. Surprisingly, we found that PPR was significantly higher in the Prosapip1 cKO group as compared to control mice

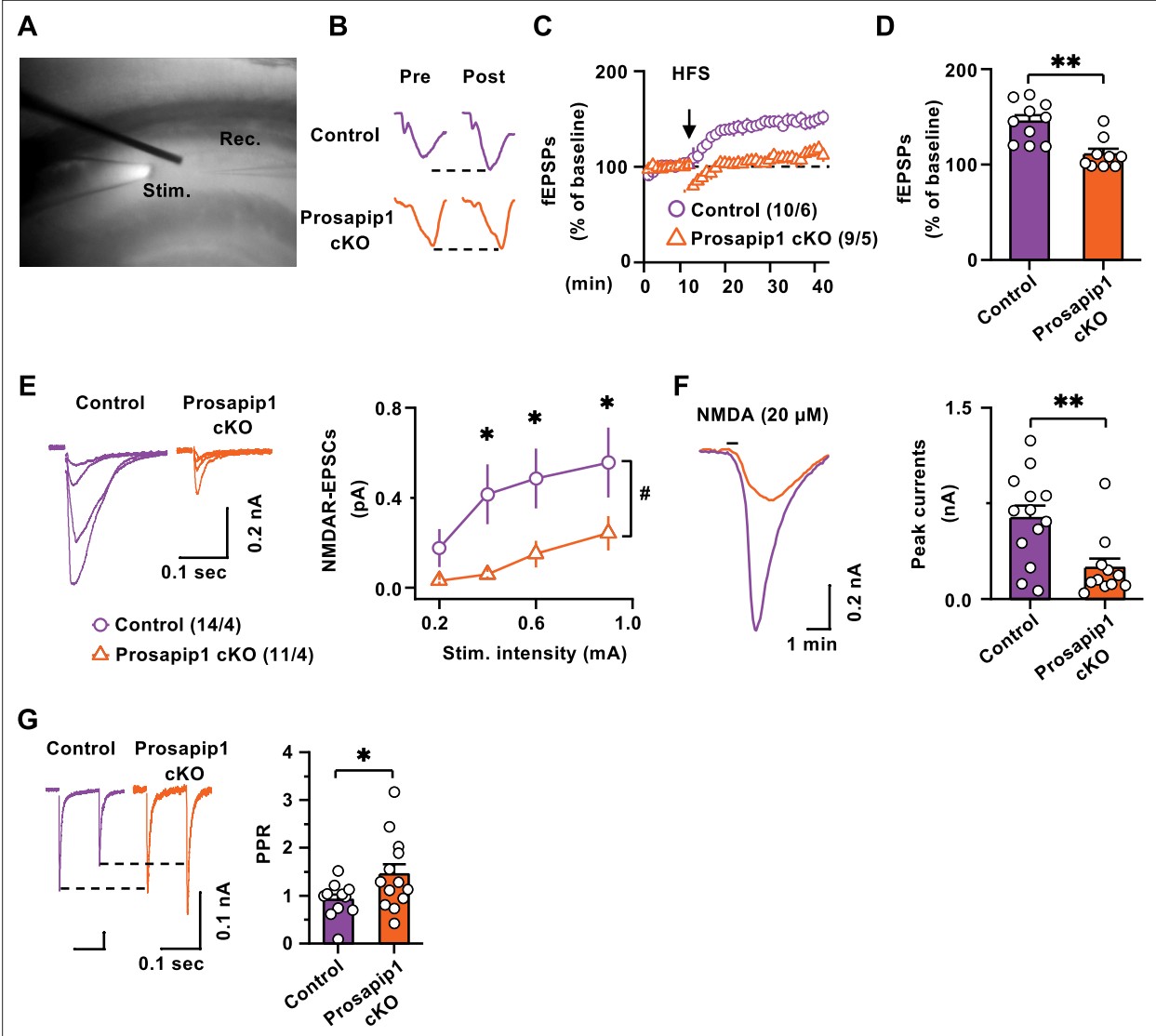

**Figure 3.** Prosapip1 in the dHP plays a role in NMDA receptor-mediated transmission and long-term potentiation. (**A**) Location of stimulating and recording electrodes within the hippocampal CA1 region. (**B**) Sample of field excitatory postsynaptic potential (fEPSP) traces recorded before (pre) and after (post) administering high-frequency stimulation (HFS) (100 Hz, 100 pulses every 20 s) in Prosapip1 cKO (Cre(+)) and control (Cre(-)) mice. (**C**) A stable baseline of fEPSPs was established for 10 min before application of HFS and fEPSPs were recorded for 30 min after HFS. Time course of fEPSPs before and after HFS. (**D**) Quantification of average fEPSP amplitudes measured between 30–40 min. Data are represented as mean ± SEM and analyzed using unpaired two-tailed t-test (**Table 1**). **p<0.01. n=10 slices from 6 mice (10/6) (Cre(-)) and (9/5) (Cre(+)). (**E**) Cells were clamped at –65 mV and the bath contained both DNQX and PTX to block AMPA and GABA-mediated responses. Voltage clamp whole cell recordings and representative electrically evoked NMDA currents in control (Cre(-)) and Prosapip1 cKO (Cre(+)) mice at four stimulation intensities (left). Summarized responses of control (Cre(-)) and Prosapip1 cKO (Cre(+)) CA1 neurons quantified by average at each stimulating intensity; #p<0.05 by two-way repeated measures ANOVA followed by post-hoc Tukey test (Cre(-) vs. Cre(+)) at the same stimulating intensities, *p<0.05. n=14/4 (Cre(-)) and 11/4 (Cre(+)). (**F**) Representative currents evoked in the CA1 neurons after NMDA bath application (20 μM, 30 s) in control (Cre(-)) and Prosapip1 cKO (Cre(+)) mice (left). Average of the peak current elicited by each mouse (right). Data are represented as mean ± SEM and analyzed using unpaired two-tailed t-test (**Table 1**). **p<0.01. n=13/4 (Cre(-)) and 11/4 (Cre(+)). (**G**) In voltage clamp recordings, two electrical stimulations (100 ms separation) were provided to elicit two responses in the CA1 neurons. Paired-pulse ratio (PPR) was calculated as the amplitude of peak 2/amplitude of peak 1. Representative paired-pulse ratio in control (Cre(-)) and Prosapip1 cKO (Cre(+)) mice (left). Average PPR for control (Cre(-)) and Prosapip1 cKO (Cre(+)) (right). Data are represented as mean ± SEM and analyzed using unpaired two-tailed t-test (**Table 1**). *p<0.05. n=11/3 (Cre(-)) and 13/4 (Cre(+)).

The online version of this article includes the following source data for figure 3:

**Source data 1.** File containing raw data for **Figure 3**.

(*Figure 3G*, *Table 1*), indicating a probable reduction in presynaptic glutamate release onto CA1 neurons in the Prosapip1 cKO mice (*Zucker and Regehr, 2002*). Together, the data imply that the Prosapip1 is crucial for the induction of hippocampal LTP, which is likely attributed to the downregulation of NMDAR function.

## Prosapip1 contributes to spatial memory

Prosapip1 was originally identified in hippocampal neurons (*Wendholt et al., 2006*; *Reim et al., 2016*). The hippocampus is a brain region associated with memory (*Broadbent et al., 2004*). Given the altered PSD composition and subsequent loss of LTP in this region after Prosapip1 knockout, we hypothesized that Prosapip1 contributes to memory-dependent behavior. First, we tested whether Prosapip1 contributes to long-term recognition memory in male and female mice using the novel object recognition (NOR) test with an inter-trial interval of 24 hr (*Leger et al., 2013*; *Lueptow, 2017*). Control animals showed a significant preference for the novel object, indicating long-term memory of the familiar object and its spatial location (*Figure 4A*, *Table 1*). Contrarily, male and female Prosapip1 cKO mice spent a similar amount of time exploring both objects, performing at chance levels (*Figure 4A*, *Table 1*). The lack of novel object recognition in Prosapip1 knockout mice implies a loss of object characteristic acquisition or long-term recognition memory.

To determine whether Prosapip1 contributes to spatial learning or working memory, male and female mice performed a novelty T-maze test (d'*d'Isa et al., 2021*). Mice were allowed to explore two arms of a 'T'-shaped maze during training trials while the third arm was blocked. All three arms were available during the test. The experiment utilized a 1 min inter-trial interval, performance of which is reliant on hippocampal LTP (*Sanderson et al., 2009*). During the test trial, control mice primarily explored the novel arm of the T-maze, indicated by a positive difference score (time spent in novel arm – time spent in familiar arm) (*Figure 4B*, *Table 1*). Prosapip1 cKO mice explored the arms at comparable levels, performing significantly worse than control animals, and indicating a lack of spatial learning for the familiar arm of the maze (*Figure 4B*, *Table 1*). This difference can be seen visually in the heatmap (*Figure 4C*). Taken together, these data suggest that Prosapip1 is required for spatial learning and working memory.

We next utilized the 3-chamber social interaction (3CSI) test. The 3CSI test has two stages. The first part measures the preference for social interaction with a juvenile interaction partner. We found that male and female Prosapip1 cKO and control mice behave similarly, with both significantly preferring the interaction partner over the empty chamber (*Figure 4D*, *Table 1*). In the second stage of the 3CSI, the mouse has the choice between interacting with a familiar partner (partner from part 1) or a novel interaction partner. This assesses the social novelty recognition of the subject. Control mice displayed a preference for the novel interaction partner, exhibited by the increase in time spent in close proximity to the novel partner (*Figure 4E*, *Table 1*). Prosapip1 cKO mice spent a similar amount of time with the familiar and novel interaction partners (*Figure 4E*, *Table 1*). The lack of discrimination from Prosapip1 cKO mice implies a loss of recognition of social novelty or loss of social memory.

While the Prosapip1 cKO mice show impaired performance in the NOR and novelty T-maze tests, these results may be attributed to the learning/acquisition process or storage of memory. To confirm that Prosapip1 is important for processes underlying spatial learning and memory, we employed the Barnes maze (*Pitts, 2018*). The Barnes maze is a white plastic circle with holes evenly drilled around the perimeter. Underneath one of these holes is an exit tunnel. The protocol has a training and testing phase, where spatial learning and memory, respectively, are separately examined. First, mice were habituated to the platform. There were no differences in baseline exploratory profile between Prosapip1 cKO and control mice (*Figure 4—figure supplement 1A*, *Table 1*). The primary variable examined during training trials was the path to escape, which tracked the distance traveled from the starting point to the exit compartment. A shorter path to escape is more efficient and indicates learning of the paradigm objective and exit location. Control mice immediately improve performance, with a significant decrease in path to escape observed as soon as the third trial (*Figure 4F*, *Table 1*). Prosapip1 cKO mice improved performance over time but more gradually and performed significantly worse than control mice over the entirety of training (*Figure 4F*, *Table 1*). Control mice showed a significantly reduced path to escape by the second trial (*Figure 4F*, *Table 1*). In addition, Prosapip1 cKO mice committed significantly more primary (incorrect hole visit) and secondary (incorrect hole revisit) errors throughout training (*Figure 4G*, *Table 1*).

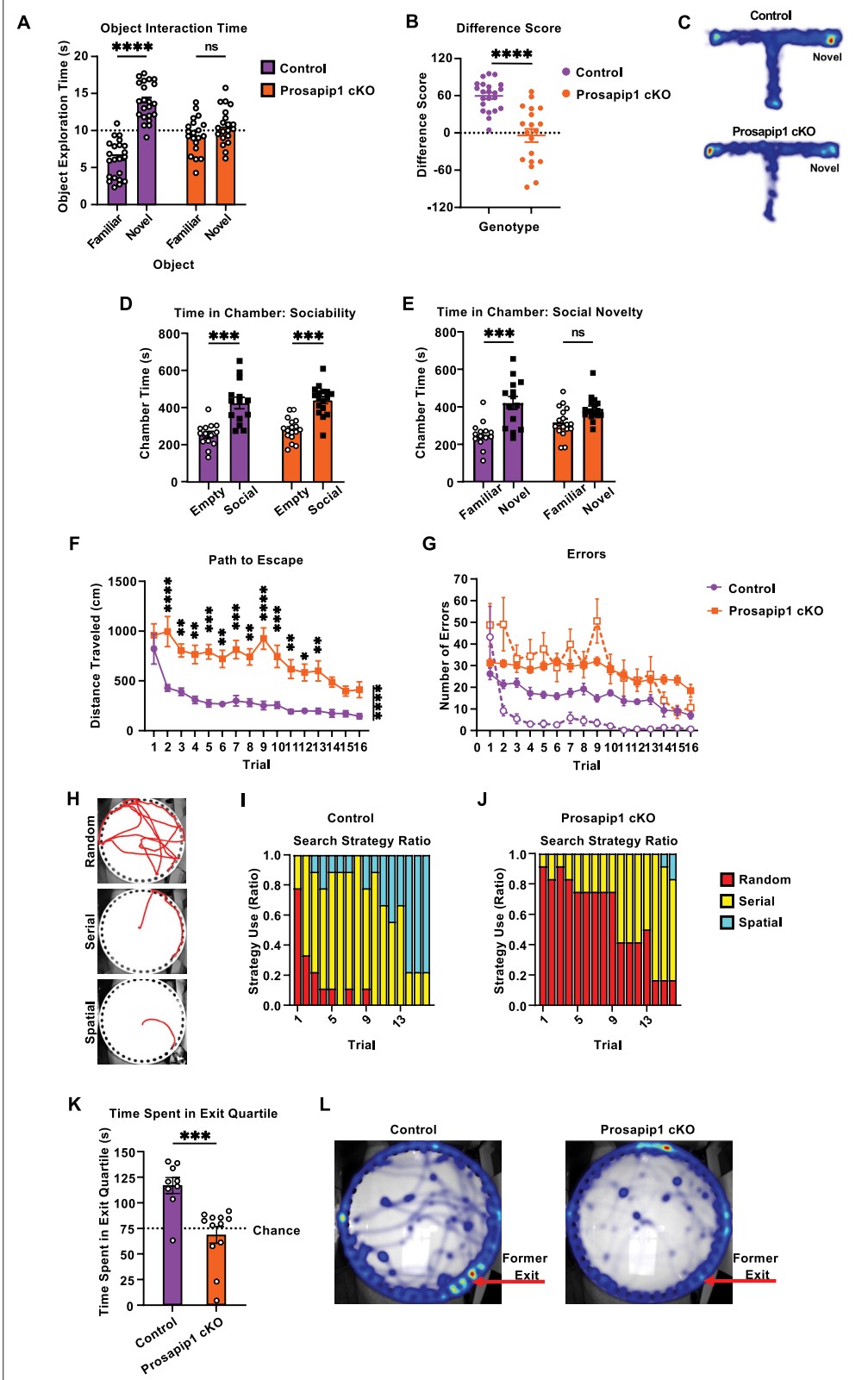

**Figure 4.** Prosapip1 contributes to recognition, social, and spatial memory. (**A**) Mice underwent the novel object recognition test, where they were first allowed to explore two similar objects. After 24 hr, one of the familiar objects was replaced by a novel object, and mice were again allowed to explore and interact with the objects. Time spent interacting with the familiar and novel objects. n=22 (control), 20 (Prosapip1 cKO). (**B–C**) In the novelty

*Figure 4 continued on next page*

*Figure 4 continued*

T-maze test, mice were allowed to explore two arms of a three-armed, T-shaped maze. There were five training trials separated by a 1 min inter-trial interval. During testing, the third 'novel' arm was unblocked and allowed to be explored. (**B**) Difference score (time exploring novel arm – time exploring familiar arm) performance during the novelty T-maze test. A positive difference score indicates preference for the novel arm of the maze. n=22 (control), 19 (Prosapip1 cKO). (**C**) Heat map of group average time spent in each arm of the T-maze during the test trial. (**D–E**) Mice performed the 3-chamber social interaction test. Specifically, they were placed in the center chamber of a 3-chamber apparatus and allowed to freely explore for 15 min for two trials. During the first trial, one chamber was paired with a juvenile interaction partner (social), while the other chamber contained only the empty interaction cage (empty). During the second trial, one chamber was paired with the familiar mouse from the first trial (familiar), and the other chamber contained a novel juvenile interaction partner (novel). (**D**) Time spent in the empty and social-paired chamber, respectively, in the first portion of the 3-chamber social interaction test. n=14 (control), 17 (Prosapip1 cKO). (**E**) Time spent in the familiar and novel chamber during the second portion of the 3-chamber social interaction test. n=14 (control), 17 (Prosapip1 cKO). (**F–K**) Mice performed the Barnes maze test, where they were placed in the center of a white plastic platform with 40 uniformly distributed holes around the perimeter, one of which had an exit compartment placed underneath. The goal of the trial was to escape into the exit compartment. There were four training trials a day over the course of 4 days, separated by an inter-trial interval of 30 min. 24 hr after the last training trial, mice were placed back onto the platform but with the exit compartment removed (probe trial) and allowed to explore for 5 min. (**F**) Average distance traveled from start point to exit during the Barnes maze training trials. n=9 (control), 12 (Prosapip1 cKO). (**G**) Primary (filled circles) and secondary (hollow circles) errors committed during Barnes maze training. Primary errors are an incorrect hole visit and secondary errors are an incorrect hole revisit. n=9 (control), 12 (Prosapip1 cKO). (**H**) The method of searching utilized by each mouse for each training trial was qualified. Example path to exit from mice exhibiting random, serial, and spatial search strategies. (**I–J**) Ratio of search strategy utilization by control (**I**) and Prosapip1 cKO (**J**) mice during Barnes maze training. (**K–L**) Time spent in exit-associated quartile during the probe trial and associated heatmaps (**L**). n=9 (control), 12 (Prosapip1 cKO). Data are represented as mean ± SEM and analyzed using two-way ANOVA (**A, C, D, E**), Welch's t-test (**B**), three-way ANOVA (**F**), or Mann-Whitney test (**J**) (*Table 1*). *p<0.05, **p<0.01, ***p<0.001, ****p<0.0001.

The online version of this article includes the following source data and figure supplement(s) for figure 4:

**Source data 1.** File containing raw data for *Figure 4*.

**Figure supplement 1.** Prosapip1 knockout did not affect Barnes maze exploratory behavior or baseline anxiety.

**Figure supplement 1—source data 1.** File containing raw data for *Figure 4—figure supplement 1*.

Differences in training performance can be explained by examining the search strategy of each mouse. There are three defined search strategies: random, serial, and spatial, which correspond to an increase in task efficiency (*Figure 4H*; *Rosenfeld and Ferguson, 2014*). Mice searching serially methodically inspected holes with minimal directional changes (*Rosenfeld and Ferguson, 2014*). Spatial searchers used environmental cues to navigate efficiently to the exit with less than 10 primary errors (*Rosenfeld and Ferguson, 2014*). In contrast, random search strategies involved frequent direction changes and errors, with mice often skipping between holes (*Rosenfeld and Ferguson, 2014*). Control mice mainly searched randomly during the first trial, but 66% of mice utilized a serial search strategy by the second trial (*Figure 4I*). Spatial searching appeared as early as trial 3, and its ratio increased until it had become a majority on the last day of training, with 77% of mice spatially searching on the final trial (*Figure 4I*). In the Prosapip1 cKO mice, randomly searching was a prevalent strategy throughout training, with 39% of all trials being classified as random (*Figure 4J*, *Table 1*). More than 50% of Prosapip1 cKO mice switched to serial search strategy by the third day of training, which also explains the gradual decrease in path to escape (*Figure 4J*, *Table 1*). Interestingly, less than 2% of all trials completed by Prosapip1 cKO mice utilized the spatial search strategy (*Figure 4J*, *Table 1*). Both groups remember the context of the Barnes maze and the objective to escape, but Prosapip1 cKO mice fail to remember the spatial location of the exit. These results confirm the conclusion that Prosapip1 is required for spatial learning.

The probe trial of the Barnes maze specifically assesses the spatial memory of the mouse. During the probe trial, the exit was removed, and the mouse was allowed to explore the platform for 5 min. If the mouse retained spatial memory of the exit, it spent significantly more time exploring the quartile associated with the exit compartment compared to chance (25% of the time). Control mice performed above chance value, spending an average of 116.84 s in the exit quartile (*Figure 4K*, *Table 1*).

Prosapip1 cKO mice, conversely, performed around chance, spending an average of 68.52 s in the exit quartile (*Figure 4K*, *Table 1*). This was significantly less than control mice and can be observed in the probe trial heat map (*Figure 4L*). Thus, Prosapip1 is required for spatial memory in the Barnes maze.

To exclude the possibility that Prosapip1 cKO mice preferred familiar situations compared to novel situations due to an increase in anxiety, we used the light/dark box and elevated plus maze paradigms to assess the baseline anxiety of these mice. The light/dark box consists of two chambers where the subject may choose to explore the well-lit ('light') or visible-light-blocked ('dark') side of a chamber (*Bourin and Hascoët, 2003*). Spending less time in the light side of the chamber implies an increase in anxiety (*Bourin and Hascoët, 2003*). Both Prosapip1 cKO and control mice spent significantly more time in the dark side compared to the light side, but there was no significant difference between the genotypes (*Figure 4—figure supplement 1B*, *Table 1*). Mice also underwent the elevated plus maze test. In this case, the maze is elevated above the floor and consists of two 'closed' arms with opaque walls and two 'open' arms with no railing. Time exploring the open arm is used to assess anxiety (*Walf and Frye, 2007*). Prosapip1 cKO and control mice spent a similar amount of time exploring the open arm (*Figure 4—figure supplement 1C*, *Table 1*). Taken together, our data suggest that Prosapip1 is not involved in anxiety-related behaviors.

## Prosapip1 in the dorsal hippocampus contributes to recognition, social, and spatial memory

Memory is often associated with the dHP (*Pilly and Grossberg, 2012*). As our data suggest that Prosapip1 contributes to recognition, social, and spatial memory, we next determined whether the dHP is the loci of Prosapip1's contribution to learning and memory processes. To do so, the dHP of *Lzts3*fl/fl mice was infected with an adeno-associated virus (AAV) expressing Cre to knockout Prospaip1 specifically in this region. Stereotaxic surgery was performed when mice were between 5 and 6 weeks of age, to ensure full viral expression by the 8–9 weeks of age. An AAV expressing solely GFP was used as a control (*Figure 5A*). Western blot analysis of the AAV-Cre-infected dHP shows efficient deletion of Prosapip1 protein (*Figure 5B*).

We repeated the experimental battery performed on the genetically manipulated mice on the mice infused with AAV-GFP or AAV-Cre in the dHP. In the open field, AAV-GFP- and AAV-Cre-infected mice exhibited similar levels of locomotion (*Figure 5C*, *Table 1*) and spent a similar amount of time in the center of the field (*Figure 5D*, *Table 1*). The region-specific knockout of Prosapip1 did not affect exploratory behavior. We also directly assessed anxiety-like behavior using the light/dark box. Mice infected with AAV-GFP or AAV-Cre both spent comparable amounts of time in the light and dark chambers, preferring the dark (*Figure 5E*, *Table 1*). Therefore, Prosapip1 in the dHP does not control locomotion or anxiety-like behavior.

Next, we examined the role of Prosapip1 in the dHP on memory. We first assessed recognition memory using the novel object recognition test. The AAV-GFP-infected mice showed a significant preference for the novel object, indicating functioning long-term recognition memory (*Figure 5F*, *Table 1*). Mice infected with AAV-Cre, however, showed no significant preference for either object (*Figure 5F*, *Table 1*). Similarly, in the novelty T-maze test, AAV-GFP-infected mice spent much of the test exploring the novel arm of the maze, indicated by a positive difference score (*Figure 5G*, *Table 1*). AAV-Cre-infected mice spent significantly less time exploring the novel arm of the maze, performing at chance levels (*Figure 5G*, *Table 1*). Together, these results suggest that Prosapip1 in the dHP specifically contributes to recognition and spatial working memory.

We also assessed the effects of region-specific Prosapip1 knockout on sociability and social memory using the 3-chamber social interaction test. Both AAV-GFP- and AAV-Cre-infected mice showed a significant preference for interacting with a social partner over the empty cage (*Figure 5H*, *Table 1*). When a novel interaction partner was introduced, AAV-GFP mice significantly preferred interacting with the novel partner (*Figure 5I*, *Table 1*). AAV-Cre-infected mice did not show a significant preference (*Figure 5I*, *Table 1*).

Finally, AAV-GFP- and AAV-Cre-infected mice underwent the Barnes maze procedure to separately assess spatial learning and memory. During training trials, AAV-Cre-infected mice performed significantly worse when examining path to escape (*Figure 5J*, *Table 1*). While AAV-GFP-infected mice quickly improved exit strategy, AAV-Cre-infected mice were more variable and took a longer path, on average (*Figure 5J*, *Table 1*). After training, during the probe trial, AAV-GFP-infected mice spent

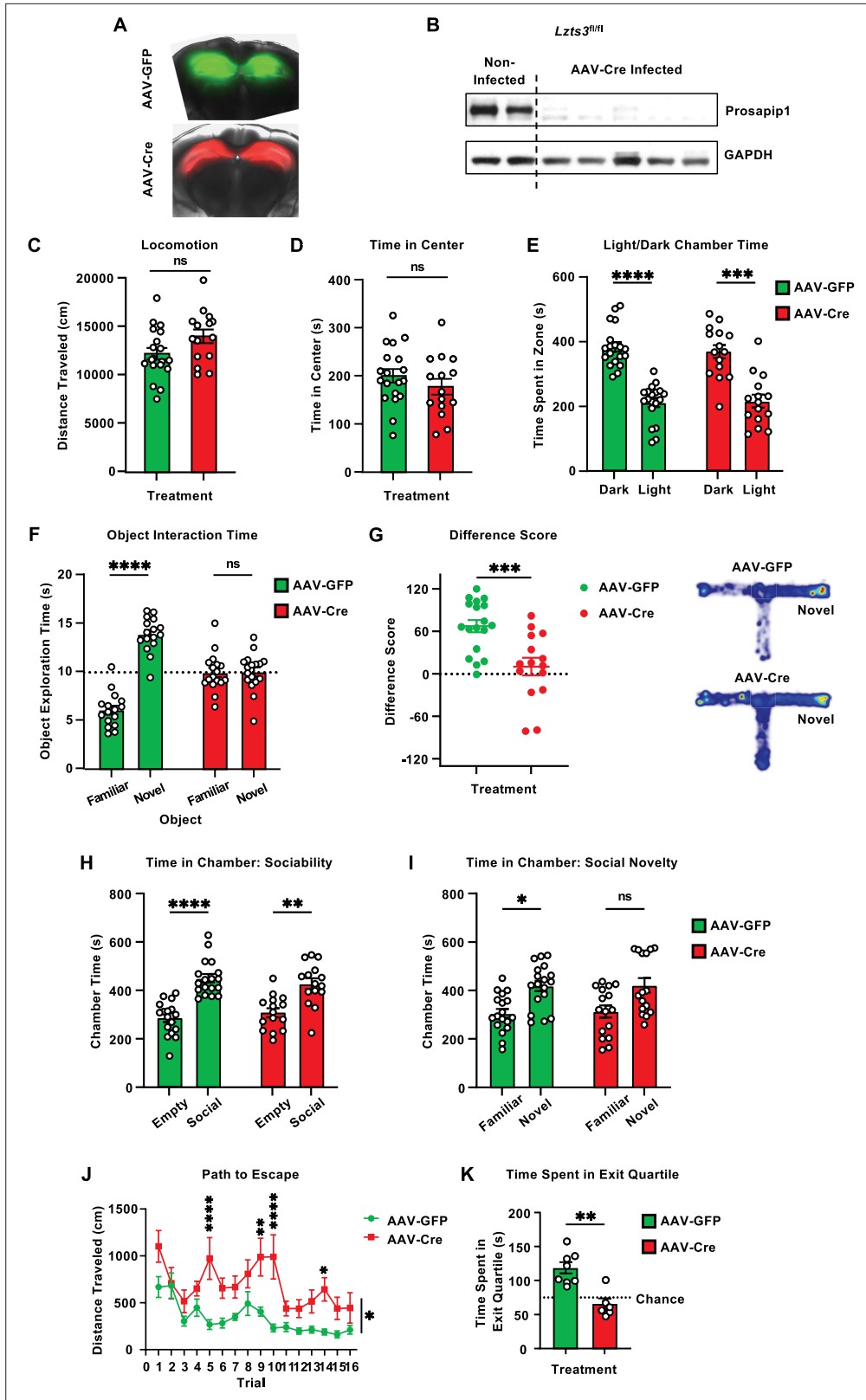

**Figure 5.** Prosapip1 in the dHP contributes to recognition, social, and spatial memory. (**A**) Images of adeno-associated virus (AAV)-GFP (green) and AAV-Cre (red) overexpression in the dHP of *Lzts3*^fl/fl mice. (**B**) Western blot analysis of Prosapip1 protein level in the dHP in non-infected mice compared to mice infected with AAV-Cre. (**C–D**) Mice infected with AAV-GFP or AAV-Cre in the dHP were placed in an open field, and their behavior was recorded

*Figure 5 continued on next page*

*Figure 5 continued*

for 20 min. The total distance traveled (**C**) and the time spent in the center of the field (**D**) were measured during the test. n=19 (AAV-GFP), 15 (AAV-Cre). (**E**) Mice infected with AAV-GFP or AAV-Cre in the dHP were placed on the light side of a light/dark box apparatus and allowed to explore for 10 min. Time spent in the dark and light chambers during the light/dark box test. n=19 (AAV-GFP), 15 (AAV-Cre). (**F**) Mice infected with AAV-GFP or AAV-Cre in the dHP underwent the novel object recognition test. Briefly, they were familiarized with two similar objects before one was switched for a novel object after 24 hr. Cumulative time spent exploring the familiar and novel object during the novel object recognition test. n=16 (AAV-GFP), 17 (AAV-Cre). (**G**) Mice infected with AAV-GFP or AAV-Cre in the dHP underwent the novelty T-maze test, where they were first allowed to explore two arms of a three-armed, T-shaped maze. During the testing phase, the third 'novel' arm was unblocked and made available for exploration. Difference score (time exploring novel arm – time exploring familiar arm) of time spent exploring the novel arm of the T-maze. A heatmap of each average group performance during the test is also presented. n=18 (AAV-GFP), 15 (AAV-Cre). (**H–I**) Mice infected with AAV-GFP or AAV-Cre in the dHP performed the 3-chamber social interaction test. Briefly, they were placed in the center chamber and allowed to freely explore for 15 min during two trials. In the first trial, one chamber had a juvenile interaction partner, and the other was empty. In the second trial, one chamber had the familiar mouse from the first trial, and the other had a new juvenile interaction partner. (**H**) Cumulative time spent in the empty and social chamber of the 3-chamber social interaction test. n=18 (AAV-GFP), 15 (AAV-Cre). (**I**) Cumulative time spent in the familiar and novel chambers in the 3-chamber social interaction test. n=18 (AAV-GFP), 15 (AAV-Cre). (**J–K**) Mice infected with AAV-GFP or AAV-Cre in the dHP underwent the Barnes maze experiment. They were placed in the center of a platform with 40 evenly spaced holes around the perimeter, one of which led to an exit compartment. Mice underwent four training trials per day for 4 days, with 30 min intervals between trials. Twenty-four hours after the last training trial, they were placed back on the platform without the exit compartment (probe trial) and allowed to explore for 5 min. (**J**) Average distance traveled from start point to exit during the Barnes maze training trials. n=8 (AAV-GFP), 6 (AAV-Cre). (**K**) Time spent in the exit quartile during the probe trial. n=8 (AAV-GFP), 6 (AAV-Cre). Data are represented as mean ± SEM and analyzed using two-way ANOVA (**E, F, H, I, J**), Welch's t-test (**C, D, G**), or Mann-Whitney test (**K**) (*Table 1*). *p<0.05, **p<0.01, ***p<0.001, ****p<0.0001.

The online version of this article includes the following source data for figure 5:

**Source data 1.** File containing raw data for *Figure 5*.

**Source data 2.** Original files for western blot analysis displayed in *Figure 5*.

---

significantly more time in the exit quartile compared to AAV-Cre-infected mice (*Figure 5K*, *Table 1*). Together, our data suggest that Prosapip1 in the dHP is specifically involved in the formation of spatial memory.

## Discussion

Our findings suggest that Prosapip1 in the dHP is responsible for the synaptic localization of SPAR and PSD-95, and consequently, for the membrane localization of GluN2B. Furthermore, our data suggest that Prosapip1 in the dHP plays an important role in LTP as well as learning and memory.

### Prosapip1 is a key protein in the PSD

Prosapip1 is highly enriched in the PSD of primary hippocampal neurons (*Wendholt et al., 2006*; *Reim et al., 2016*), and our results suggest that Prosapip1 could be crucial to the formation and stability of PSD complexes. Prosapip1 scaffolds SPAR to Shank3 in the PSD (*Wendholt et al., 2006*; *Dolnik et al., 2016*; *Reim et al., 2016*). We found that knockout of Prosapip1 reduced the synaptic localization of SPAR but not Shank3. Reim et al. reported that Prosapip1 is directly regulating SPAR levels in the PSD in primary hippocampal neurons (*Reim et al., 2016*). Our data supports this study, as Prosapip1 knockout in vivo led to a reduction of synaptic SPAR. SPAR plays a role in the formation of F-actin, which is important for dendritic spine maturation (*Pak et al., 2001*). We previously showed Prosapip1 interacts with SPAR in the NAc of mice (*Laguesse et al., 2017*), and Prosapip1 knockdown reduces F-actin content while overexpression of Prosapip1 increases F-actin content in the NAc (*Laguesse et al., 2017*). Furthermore, we showed that Prosapip1 plays a role in spine maturation in the NAc (*Laguesse et al., 2017*). Our results suggest that the recruitment of SPAR to the PSD via Prosapip1 in the dHP promotes F-actin formation and lead to dendritic spine maturation, however, these possibilities merit further examined.

Both SPAR and Shank3 interact with PSD-95 (*Naisbitt et al., 1999*; *Tu et al., 1999*; *Pak et al., 2001*). SPAR binds PSD-95 directly (*Pak et al., 2001*), and Shank3 associates with Guanylate kinase-associated protein (GKAP), which binds PSD-95 (*Naisbitt et al., 1999*; *Tu et al., 1999*). PSD-95 binds to NMDAR subunits GluN2A and GluN2B and stabilizes NMDAR surface localization (*Won et al., 2016*; *Coley and Gao, 2019*). Studies have shown that PSD-95 deficiency causes an imbalance of NMDAR and AMPAR synaptic presence, altering glutamatergic transmission (*Lin et al., 2004*; *Elias et al., 2006*; *Zhang et al., 2013*; *Chen et al., 2015*). Surprisingly, we discovered that loss of Prosapip1 resulted in a marked reduction of synaptic PSD-95, and a correlated reduction in the synaptic compartmentalization of the NMDAR subunit GluN2B, raising the possibility that the interaction between Prosapip1 and PSD-95 is required for synaptic membrane localization of GluN2B. However, we did not observe a change in the synaptic level of GluN2A. In addition, we did not find that the localization of the AMPAR subunit GluA1 was altered as a result of Prosapip1 knockout. In line with our findings, *Pak et al., 2001* showed that SPAR binds PSD-95 and GluN2B, but not GluA1. Based on our results and these studies, the complex controlling AMPAR synaptic localization is different, and likely independent of Prosapip1 in the dHP. Together, our data suggest that Prosapip1 is necessary for the formation and/or maintenance of the NMDAR-associated PSD network.

We previously discovered that Prosapip1's translation depends on mTORC1 in the NAc of mice consuming excessive levels of alcohol, which in turn promotes F-actin formation, dendritic spine maturation, and alcohol-reward memory and the reinforcement of alcohol consumption (*Laguesse et al., 2017*). mTORC1 activation by alcohol also induces the translation of the scaffolding protein Homer and PSD-95 in the NAc of mice (*Beckley et al., 2016*; *Liu et al., 2017*). mTORC1 plays an essential role in the translation of a subset of mRNAs to proteins and in the CNS, mTORC1 promotes the translation of synaptic proteins at dendrites (*Buffington et al., 2014*), and plays an important role in LTP, learning, and memory (*Hoeffer and Klann, 2010*; *Graber et al., 2013*). Therefore, it would be of great interest to test the hypothesis that learning-dependent activation of mTORC1 in the dHP activates the translational machinery at synapses and, therefore, increases the translation of Prosapip1, which in turn recruits proteins such as PSD-95 to stabilize the PSD-95 complex.

## Prosapip1 is required for LTP in the dHP

In our study, we discovered that mice lacking Prosapip1 exhibited markedly diminished NMDAR-mediated synaptic transmission, which is in line with the fact that the synaptic GluN2B levels are reduced. We also found that Prosapip1 knockout led to a significant reduction in the formation of LTP. LTP is a persistent increase in synaptic strength between neurons that can last for months in vivo (*Abraham et al., 2002*; *Cooke and Bliss, 2006*) and is the cellular hallmark of learning and memory. It is often induced by HFS, which causes extensive glutamate release and significant AMPAR activity, resulting in large membrane depolarization (*Ma et al., 2018*). This depolarization facilitates the removal of $Mg^{2+}$ blockage in NMDARs, thereby activating them and leading to calcium influx (*Ma et al., 2018*). This influx triggers the calmodulin/CaMKII pathway, initiating a series of cascades that enhance AMPAR phosphorylation and trafficking, culminating in sustained synaptic strengthening (*Huganir and Nicoll, 2013*; *Herring and Nicoll, 2016*). Therefore, the induction of hippocampal LTP heavily relies on NMDAR activation, with alterations in NMDAR function significantly affecting the establishment of hippocampal LTP and its associated learning and memory processes. In Prosapip1 knockout mice, the reduction in NMDAR functionality likely results in decreased calcium influx during high-frequency stimulation and diminished activation of the calmodulin/CaMKII pathway, leading to weaker synaptic strengthening.

## Prosapip1 is required for learning and memory

As detailed above, we found that Prosapip1 neuronal knockout mice have deficits in LTP in the dHP, and LTP is the hallmark of learning and memory (*Bliss and Collingridge, 1993*; *Frey and Morris, 1998*; *Collingridge et al., 2004*). We also found that global neuronal knockout and dHP-specific knockout of Prosapip1 results in deficits in spatial learning and memory. Specifically, we showed that Prosapip1 in the dHP is required for normal performance in the novel object recognition, novelty T-maze, 3-chamber social interaction, and Barnes maze tests, which all require a unique form of learning and memory (*Davis et al., 1992*; *Broadbent et al., 2010*; *Barker and Warburton, 2011*; *Rosenfeld and Ferguson, 2014*; *Lueptow, 2017*; *Pitts, 2018*; *Sánchez-Rodríguez et al.*,

*2022*). The novel object recognition test is primarily examining recognition memory, and with an inter-trial interval of 24 hr, specifically long-term memory (*Broadbent et al., 2010*; *Barker and Warburton, 2011*; *Antunes and Biala, 2012*; *Lueptow, 2017*; *Cinalli et al., 2020*). The novelty T-maze focuses on both spatial learning and memory, and with its short, 1 min inter-trial interval, this procedure is examining spatial working memory (*Sharma et al., 2010*; d'*d'Isa et al., 2021*). The 3-chamber social interaction test is assessing baseline sociability and also social recognition memory (*Meira et al., 2018*; *Tzakis and Holahan, 2019*; *Wang and Zhan, 2022*; *Cope et al., 2023*; *Wei et al., 2024*). Finally, the Barnes maze tests both spatial learning and working memory in intra-day trials, while simultaneously testing long-term contextual and spatial memory in inter-day trials and the probe test (*Bach et al., 1995*; *Sharma et al., 2010*; *Rosenfeld and Ferguson, 2014*; *Pitts, 2018*). Interestingly, loss of hippocampal LTP has been shown to impair spatial, but not contextual memory in the Barnes maze (*Bach et al., 1995*). As presented here, Prosapip1 knockout mice significantly reduced distance traveled to exit but did not switch to spatial searching. In this example, Prosapip1 knockout mice are retaining the contextual understanding of escaping the platform but do not recall the spatial location of the exit. Additionally, the link between NMDAR function and learning and memory is well established (*Newcomer et al., 2000*; *Li and Tsien, 2009*). For example, blockage of hippocampal NMDA receptors impairs spatial learning in rats (*Davis et al., 1992*; *Bye and McDonald, 2019*). Prosapip1 is therefore likely controlling the reinforcement of learning and memory by PSD scaffolding, stabilization, and GluN2B synaptic localization, leading to LTP.

We observed that Prosapip1 knockout specifically in the dHP replicated the recognition, social, and spatial learning and memory deficits exhibited by the global neuronal knockout mice, suggesting that Prosapip1 is controlling these learning and memory processes specifically in the dHP. The dHP primarily controls memory formation and recall (*Eichenbaum, 1997*; *Broadbent et al., 2004*; *Squire et al., 2004*; *Pilly and Grossberg, 2012*). We found that LTP in the CA1 subregion of the dHP was reliant on Prosapip1. The CA1 subregion is critically involved in contextual memory, object recognition memory, and spatial memory (*Tsien et al., 1996*; *Lee and Kesner, 2004*; *Daumas et al., 2005*; *Sanderson et al., 2009*; *Sharma et al., 2010*; *Stevenson et al., 2018*; *Bye and McDonald, 2019*; *Cinalli et al., 2020*; *Jeong and Singer, 2022*). Loss of Prosapip1 in CA1 is likely leading to decreased performance in the novel object recognition, novelty T-maze, and Barnes maze tests. The lack of social recognition displayed by *Prosapip1*(fl/fl);Syn1-Cre(+) mice and AAV-Cre-infected mice is likely attributed to the loss of Prosapip1 in the CA2 subregion of the dHP, which is the primary subregion controlling social recognition memory (*Meira et al., 2018*; *Tzakis and Holahan, 2019*; *Wang and Zhan, 2022*; *Cope et al., 2023*; *Wei et al., 2024*). Specifically, silencing the CA2 subregion of the dHP impairs social memory formation and consolidation (*Meira et al., 2018*). However, the CA3 and DG subregions of the dHP are also involved in spatial and contextual memory (*Broadbent et al., 2004*; *Lee and Kesner, 2004*; *Daumas et al., 2005*). As our conditional knockout strategy resulted in Prosapip1 deletion from the whole dHP, further studies are required to dissect the subregion specificity of the contribution of Prosapip1 to recognition, social, and spatial learning and memory processes.

Memory consists of three primary processes: encoding, consolidation, and retrieval (*Straube, 2012*). In this study, the defect in memory function is likely due to a failure to encode new information (*Bye and McDonald, 2019*) or consolidate this 'short-term' into 'long-term' memory (*Yang et al., 2022*). The spatial T-maze experiment utilized a short inter-trial interval of 1 min, which requires working spatial memory (*Sharma et al., 2010*), and Prosapip1 knockout mice exhibited a failure to encode new information. Similarly, the Barnes maze training trials were separated by an inter-trial interval of 30 min, but Prosapip1 knockout mice did not acquire spatial memory between training trials, nor during the longer consolidation periods between days, again implying a failure to encode spatial information or consolidate this information. The lack of synaptic localization of GluN2B is likely underlying the loss of memory encoding or consolidation (*Nachtigall et al., 2024*). It is unlikely that NMDAR dysfunction is affecting the retrieval of memory, as studies have exhibited rats' ability to use previously acquired spatial information during NMDAR blockage (*Bast et al., 2005*; *Mackes and Willner, 2006*; *Bye and McDonald, 2019*).

Prosapip1 belongs to the Fezzin family of proteins (*Wendholt et al., 2006*). It is important to note that other Fezzins do not compensate for the loss of Prosapip1 in the dHP. Knockout of other Fezzins, like PSD-Zip70, also leads to cognitive deficits (*Mayanagi et al., 2015*). However, these deficits were

attributed to the action of PSD-Zip70 in the PFC. Therefore, one could hypothesize that proteins in this family enact their function in specific brain subregions.

In summary, Prosapip1 in the dorsal hippocampus is integral to the synaptic localization of SPAR, PSD-95, and GluN2B, which are required for the formation of LTP and subsequent spatial learning and memory behavior. Abnormalities with PSD proteins are associated with neuropsychiatric disorders (*Kaizuka and Takumi, 2018*), and further unraveling of the physiological role of Prosapip1 may unlock insights into normal and abnormal mechanisms of learning and memory.

# Materials and methods

## Key resources table

| Reagent type (species) or resource | Designation | Source or reference | Identifiers | Additional information |
|---|---|---|---|---|
| Biological Sample (*Mus musculus*) | Single cell C57BL/6 J embryos | Jackson Laboratory | | Electroporated with gRNA to create new mouse strain |
| Strain, strain background (*Mus musculus*, female) | CD-1 IGS Mouse | Charles River | Strain Code 022 | Pseudopregnant recipient female |
| Strain, strain background (*Mus musculus*, female) | C57BL/6 J | Jackson Laboratory | 000664 RRID:IMSR_JAX:000664 | Mating with first *Lzts3*fl/fl offspring |
| Strain, strain background (*Mus musculus*, male and female) | B6.Cg-Tg(Syn1-cre)671Jxm/J | Jackson Laboratory | 003966 RRID:IMSR_JAX:003966 | Mouse line |
| Strain, strain background (*Mus musculus*, male and female) | *Lzts3*fl/fl | First described in this paper | | Mouse line. Available upon request to Dorit Ron |
| Strain, strain background (*Mus musculus*, male and female) | *Lzts3*fl/fl;Syn1-Cre | First described in this paper | | Mouse line. Syn1-Cre available at Jackson Laboratory |
| Sequence-based reagent | Alt-R CRISPR-Cas9 crRNAs | IDT DNA | This paper | Generated by Gregg E. Homanics and available upon request. |
| Sequence-based reagent | long PAGE-purified Ultramer single-stranded DNA oligos | IDT DNA | This paper | 140 nt, target loci in intron 2 and exon 5 Generated by Gregg E. Homanics and available upon request. |
| Sequence-based reagent | Intron 2 lox insertion F | This paper (materials and methods section) | PCR primers | AGAGAAGTCTACGCTGTAGTCAG Generated by Gregg E. Homanics and available upon request. |
| Sequence-based reagent | Intron 2 lox insertion R | This paper (materials and methods section) | PCR primers | AAGCGGGAAGGTAGAGAGGT Generated by Gregg E. Homanics and available upon request. |
| Sequence-based reagent | Exon 5 lox insertion F | This paper (materials and methods section) | PCR primers | TGCACAACCTTCTGACACGT Generated by Gregg E. Homanics and available upon request. |
| Sequence-based reagent | Exon 5 lox insertion R | This paper (materials and methods section) | PCR primers | AGGGCACAGACAGTAGCACT Generated by Gregg E. Homanics and available upon request. |
| Transfected construct (*M. musculus*) | AAV8-Ef1a-mCherry-IRES-Cre | Addgene | 55632-AAV8 | $1\times10^{13}$ vg/ml. AAV to infect mouse brain. |
| Transfected construct (*M. musculus*) | AAV2-CMV-EGFP | Addgene | 105530-AAV2 | $1\times10^{13}$ vg/ml, AAV to infect mouse brain |

*Continued on next page*

*Continued*

| Reagent type (species) or resource | Designation | Source or reference | Identifiers | Additional information |
|---|---|---|---|---|
| Antibody | anti-SPAR (SIPA1L1) (Rabbit, polyclonal) | ProteinTech | 25086–1-AP RRID:AB_2714023 | WB (1:500) |
| Antibody | anti-Prosapip1 (Rabbit, polyclonal) | ProteinTech | 24936-1-AP RRID:AB_2714022 | WB (1:2000) |
| Antibody | anti-Shank3 (Mouse, monoclonal) | Abcam | Ab93607 RRID:AB_10563849 | WB (1:500) |
| Antibody | anti-GluN2B (Rabbit, monoclonal) | Cell Signaling | 4212 RRID:AB_2112463 | WB (1:1000) |
| Antibody | anti-GluA1 (Rabbit, monoclonal) | Cell Signaling | 13185 S RRID:AB_2732897 | WB (1:1000) |
| Antibody | anti-CREB (Rabbit, monoclonal) | Cell Signaling | 9197 RRID:AB_2800317 | WB (1:500) |
| Antibody | anti-GluN2A (Goat, polyclonal) | Santa Cruz | SC-1468 RRID:AB_2630886 | WB (1:500) |
| Antibody | anti-GAPDH (Mouse, monoclonal) | Sigma-Aldrich | G8795 RRID:AB_1078991 | WB (1:10,000) |
| Antibody | anti-PSD-95 (Mouse) | Millipore | 05–494 RRID:AB_2315219 | WB (1:100,000) |
| Antibody | anti-rabbit horseradish peroxidase (Donkey, polyclonal) | Jackson ImmunoResearch | 711-035-152 RRID:AB_10015282 | WB (1:5,000) |
| Antibody | anti-goat HRP (Donkey, polyclonal) | Jackson ImmunoResearch | 705-035-003 RRID:AB_2340390 | WB (1:5,000) |
| Antibody | anti-mouse HRP (Donkey, polyclonal) | Jackson ImmunoResearch | 715-035-150 RRID:AB_2340770 | WB (1:5,000) |
| Commercial assay or kit | Pierce bicinchoninic acid (BCA) protein assay kit | Thermo Scientific | 23225 | |
| Commercial assay or kit | NuPAGE Bis-Tris precast gels | Life Technologies | NP00321BOX | |
| Chemical compound, drug | Enhance Chemiluminescence reagent (ECL) | Millipore | WBAVDCH01 | |
| Chemical compound, drug | cOmplete, Mini, EDTA-free Protease Inhibitor Cocktail | Roche | 11836170001 | |

## Ethics statement

All animal procedures were approved by UCSF Institutional Animal Care and Use Committee (IACUC) (animal protocol AN206967) and were conducted in agreement with the Association for Assessment and Accreditation of Laboratory Animal Care (AAALAC).

## Methods of Euthanasia

Mice were euthanized by carbon dioxide inhalation followed by cervical dislocation, or deep anesthetization with pentobarbital (150 mg/kg) followed by transcranial perfusion with 4% paraformaldehyde. These procedures are in accordance with the Panel on Euthanasia of the American Veterinary Medical Association guidelines and with the standard operating procedures of the UCSF IACUC.

## Animals

*Lzts3*[fl/fl] mice were generated as described below. Syn1-Cre(+) mice were purchased from Jackson Laboratory (Stock #003966, Bar Harbor, Maine). *Lzts3*[fl/fl] and *Lzts3*[fl/fl]; Syn1-Cre mice were bred and group housed in a 12 hr light-dark cycle room that was temperature- and humidity-controlled. Unrestricted amounts of food and water were provided.

## Generation of *Prosapip1* floxed mice

Guide RNA binding sites in intron 2 and in the 3'UTR located in Exon 5 of *Lzts3* (*Figure 1A*) were identified using the CRISPRator tool (*Labuhn et al., 2018*) within the CCTop (*Stemmer et al., 2015*) online platform (http://crispr.cos.uni-heidelberg.de/). The gRNA binding sites are located ~3.3 kb from each other. These gRNA target sites were used to produce two Alt-R CRISPR-Cas9 crRNAs (IDT DNA, Coralville, IA) which were individually hybridized to a universal 67-mer Alt-R CRISPR-Cas9 tracrRNA (IDT DNA) to produce two gRNAs. Two 140 nt long PAGE-purified Ultramer single-stranded DNA oligos (IDT DNA) with three phosphorothioate modifications on each end (*Renaud et al., 2016*) that were homologous to the target loci in intron 2 and exon 5 (*Figure 1B*) were used as a repair templates.

We introduced each loxP site into 1 cell and 2 cell stages of embryonic development (*Horii et al., 2017*). Single cell C57BL/6 J embryos were electroporated with gRNA#6 (200ng/µl), IDT Alt-R HiFi Cas9 Nuclease V3 protein (75 ng/µl), and Exon 5 repair template (200 ng/µl) using a BioRad Gene-Pulser Xcell electroporator in a 1 mm-gap slide electrode (Protech International, #501P1-10) using square-wave pulses (five repeats of 3 ms 25 V pulses with 100 ms interpulse intervals). Following this first electroporation, embryos were cultured overnight. Surviving two-cell embryos were electroporated with gRNA#7 (200ng/µl), IDT Alt-R HiFi Cas9 Nuclease V3 protein (75 ng/µl), and intron 2 repair template (200 ng/µl) under conditions described above. Surviving embryos were transferred to the oviducts of CD1 (Charles River) pseudopregnant recipient females. Offspring were genotyped for the intron 2 lox insertion using PCR (forward primer 5' AGAGAAGTCTACGCTGTAGTCAG 3' and reverse primer 5' AAGCGGGAAGGTAGAGAGGT 3'; wild-type product = 449 bp, floxed product = 489 bp) followed by Sanger sequencing. Offspring were genotyped for the Exon 5 lox insertion using PCR (forward primer 5' TGCACAACCTTCTGACACGT 3' and reverse primer 5' AGGGCACAGACAGTAG CACT 3'; wild-type product = 408 bp, floxed product = 448 bp) followed by Sanger sequencing. Two founder animals were found that harbored loxP insertions at both the intron 2 and exon 5 sites. When mated to C57BL/6 J females, the loxP sites segregated, indicating that they were on different chromosomes. Therefore, F1 offspring that harbored only the loxP insertion at the intron 2 site were used as 1 cell embryo donors for insertion of the Exon 5 loxP site using electroporation conditions described above. One male offspring was produced that harbored both the intron 2 and exon 5 loxP insertions on the same chromosome. This male was mated to C57BL/6 J females to establish the floxed mouse line used here.

Guide RNA off-target sites were predicted and ranked using CRISPOR (*Concordet and Haeussler, 2018*). The top 11 and 10 sites for gRNA#6 and gRNA#7, respectively, based on CFD score were amplified from this founder mouse DNA and Sanger sequenced. All predicted off-target sites analyzed were wild-type (data not shown).

## Reagents

Antibodies: Rabbit anti-Prosapip1 (1:2000) (24936-1-AP) and Rabbit anti-SPAR (SIPA1L1) (1:500) (25086–1-AP) antibodies were purchased from ProteinTech. Mouse anti-Shank3 antibodies (1:500) (#ab93607) were purchased from Abcam. Rabbit anti-GluN2B (1:1000) (#4212), Rabbit anti-GluA1 (1:1000) (#13185 S), and Rabbit anti-CREB (1:500) (#9197) antibodies were purchased from Cell Signaling. Goat anti-GluN2A antibodies (1:500) (#SC-1468) were purchased from Santa Cruz. Mouse anti-GAPDH antibodies (1:10,000) (#G8795) were purchased from Sigma. Mouse anti-PSD-95 antibodies (1:100,000) (#05–494) were purchased from Millipore (Upstate). Donkey anti-rabbit horseradish peroxidase (HRP), donkey anti-goat HRP, and donkey anti-mouse HRP conjugated secondary antibodies (1:5000) were purchased from Jackson ImmunoResearch (West Grove, PA). Enhance Chemiluminescence reagent (ECL) was purchased from Millipore (Burlington, MA). EDTA-free complete mini–Protease Inhibitor Cocktails was from Roche (Indianapolis, IN). Pierce bicinchoninic acid (BCA) protein assay kit was purchased from Thermo Scientific (Rockford, IL). NuPAGE Bis-Tris precast gels were purchased from Life Technologies (Carlsbad, CA).

Viruses: AAV8-Ef1a-mCherry-IRES-Cre ($1\times10^{13}$ vg/ml #55632) and AAV2-CMV-EGFP ($110^{13}$ vg/ml, #105530) were purchased from Addgene.

## Crude synaptosomal fractionation

Tissue was first homogenized at 4 °C in 500 µl of Krebs buffer including 125 mM NaCl, 1.2 mM KCl, 1.2 mM $MgSO_4$, 1.2 mM $CaCl_2$, 22 mM $Na_2CO_3$, 1 mM $NaH_2PO_4$, 10 mM Glucose, 0.32 M sucrose, and protease/phosphatase inhibitors. A portion of the homogenate (100 µl) was saved as total homogenate (H), while the remaining homogenate was diluted by adding 500 µl of Krebs buffer to the remaining 400 µl. The total homogenate was transferred to a 1.5 ml Eppendorf tube. The glass homogenizer was washed with 500 µl of Krebs buffer, and the wash solution was added to the remaining homogenate. The sample was then centrifuged at 1,000 g at 4 °C for 10 min. The supernatant (S1) was collected. The process was repeated, and then the S1 supernatant was centrifuged at 16,000 g at 4 °C for 20 min. The supernatant (S2) was saved, and the pellet was kept on ice. The resulting pellet contained the synaptosomal fraction. The pellet was then resuspended in 500 µl of Krebs buffer and centrifuged at 16,000 g, 4 °C for 20 min and resuspended in radio immunoprecipitation assay (RIPA) buffer (containing 50 mM Tris-HCL, 5 mM EDTA, 120 mM NaCl, 1% NP-40, 0.1% deoxycholate, 0.5% SDS, and protease and phosphatase inhibitors) for analysis.

## Western blot analysis

Tissue collected from mice was homogenized in ice-cold RIPA buffer using a sonic dismembrator. Protein concentration was determined using the BCA protein assay kit. 30 µg of each tissue lysate was loaded for separation by SDS-PAGE (4–12%), followed by transfer onto a nitrocellulose membrane at 300 mA for 2 hr. The membranes were then blocked with 5% milk-PBS containing 0.1% Tween 20 at room temperature for 30 min before being probed with the appropriate primary antibodies overnight at 4 °C. Following washing, the membranes were incubated with HRP-conjugated secondary antibodies for 1 hr at room temperature and then visualized using ECL. Band intensities were quantified using ImageJ software (NIH, https://imagej.net/ij/).

## Electrophysiology

Preparation of slices is outlined in *Gangal et al., 2023*. In brief, coronal sections of the hippocampus (250 µm) were cut in an ice-cold solution containing the following: 40 mM NaCl, 148.5 mM sucrose, 4.5 mM KCl, 1.25 mM $NaH_2PO_4$, 25 mM $NaHCO_3$, 0.5 mM $CaCl_2$, 7 mM $MgSO_4$, 10 mM dextrose, 1 mM sodium ascorbate, 3 mM myo-inositol, 3 mM sodium pyruvate, and saturated with a 95% $O_2$ and 5% $CO_2$. After cutting, the slices were incubated for 45 min in an external solution containing the following: 125 mM NaCl, 4.5 mM KCl, 2.5 mM $CaCl_2$, 1.3 mM $MgSO_4$, 1.25 mM $NaH_2PO_4$, 25 mM $NaHCO_3$, 15 mM sucrose, 15 mM glucose, and saturated with 95% $O_2$ and 5% $CO_2$.

Field potential recordings were conducted as detailed in previous studies (*Wang et al., 2012*). Specifically, the procedure involved the use of stimulation electrodes filled with artificial cerebrospinal fluid (aCSF) and placed in the CA1 region of the dHP. The recording electrode, filled with 1 M NaCl, was positioned in the CA1 region, approximately 100–150 µm away from the stimulating electrodes. The optimal location for the recording electrode was determined based on the site where a stable fiber volley and field potential response could be consistently observed following the delivery of a single electrical stimulation pulse (2 ms). The stimulation intensity was carefully adjusted to elicit 50% of the maximum possible response. Baseline field potentials were meticulously recorded over 10 min at 10 s intervals. The HFS protocol for LTP included a regimen of 100 Hz, with 100 pulses every 20 s, repeated four times. Following the HFS protocol, field potential recordings were extended for 30 min. LTP was quantified based on the averaged fEPSP as a percentage of the baseline, comparing the measurements at 20–30 min post-HFS between the Prosapip1 cKO and control groups. To inhibit GABAergic transmission, picrotoxin (100 µM) was applied to the bath. The measurement of fEPSP was performed using a MultiClamp 700B amplifier integrated with Clampex 10.4 software provided by Molecular Devices.

Whole-cell recordings were executed following the protocols previously established (*Wang et al., 2012*; *Ma et al., 2018*). For these recordings, a cesium-based intracellular solution was employed. This solution comprised of the following components: 119 mM CsMeSO₄, 8 mM Tetraethylammonium chloride (TEA-Cl), 15 mM HEPES, 0.6 mM ethylene glycol tetraacetic acid (EGTA), 0.3 mM Na₃GTP,

4 mM MgATP, 5 mM QX-314-Cl, and 7 mM phosphocreatine. The pH of this solution was adjusted to 7.3 using CsOH. The temperature of the recording bath was maintained at a constant 32 °C, and the perfusion speed was set between 2–3 ml/min. For the recordings, CA1 neurons were voltage-clamped at a holding potential of –70 mV. In order to measure NMDAR-mediated transmission, a low external concentration of $Mg^{2+}$ (0.05 mM) was used. This was coupled with NBQX (10 μM) and picrotoxin (100 μM) to block AMPA receptor (AMPAR)-mediated synaptic transmission and inhibitory synaptic currents. NMDA-induced currents were measured by bath-applying NMDA (20 μM) for 30 s, with holding currents recorded every 5 s.

Input-output curves for NMDAR-mediated EPSC were created by electrical stimulation of varying intensities which were delivered through electrodes strategically placed within the CA1 region. The paired-pulse ratio (PPR) was calculated by dividing the amplitude of the second electrically-evoked EPSC by that of the first, with an interval of 100ms between the two pulses. Additionally, gap-free recordings of spontaneous excitatory postsynaptic currents were conducted over a duration of 2 min.

## Behavior

All behavioral paradigms were completed between the hours of 9:00 and 18:00. Mice were tested between 8 and 20 weeks of age and were age-matched based on the experiment. The mice were moved from their housing room, currently in the 'light cycle,' to the dimly lit behavior room (10–15 lux) 30 min before the experiment and allowed to acclimate. The light/dark box, elevated plus maze, and Barnes maze experiments were done in full room light. During experiments, white noise was played through a speaker in the room at 50 dB to reduce influence from noise outside of the room. The researcher remained in the room during the trial, but a wall was placed between the arena and the experimenter to prohibit mice from seeing the researcher. Mice were handled for 3 days for 1–2 min per day before experimentation began to reduce handling anxiety. All behavioral analyses were recorded and analyzed using Noldus Ethovision XT software.

## Locomotion

Mice were placed in a 43 cm × 43 cm open field and allowed to explore for 30 min. The center point of the mouse was tracked, and the primarily dependent variable was total locomotion during the trial. Other tracked variables were average and maximum velocity, center entrances, time spent in center, and locomotion binned into 1 min periods to examine evolution of exploration of the space.

## Novel object recognition

Mice were placed in a 43 cm × 43 cm chamber with two similar objects and allowed 20 s of total familiarization time (nose point within 2 cm radius of object), or 5 total min in the chamber, whichever came first, before being returned to home cage. After 24 hr, mice were allowed to explore the test space containing one object from the familiarization trial (familiar object) and one novel object until total object interaction time of 20 s was reached. If 20 s of total interaction time was not reached before the 5 min time limit, the trial was excluded (three trials). Mice that were climbing on objects were gently but quickly moved back to the starting position.

## Novelty T-Maze

The protocol was adapted from *Sanderson et al., 2009*. Mice were placed in the 'start' arm of a T-shaped maze. The dimensions of each transparent arm were 30 cm×10 cm×20 cm (L × W × H). During training, the entrance to one arm ('novel arm,' assignment counterbalanced across groups) was blocked with clear plastic. Training consisted of five, 2 min trials in which each subject was allowed to explore the 'start' and 'familiar' arms. The training trials were separated by a 1 min inter-trial interval. After the training sessions, the plastic blocking the 'novel' arm was removed and mice were allowed to explore all arms of the maze. The test session began when the center point of the mouse left the start arm and ended when 2 min of total time was spent in the novel or familiar arm. Mice that showed anxiety-related behavior and an aversion to either arm during the test (no arm entrances, <250 cm total movement) were excluded (2 mice).

## 3-chamber social interaction

Mice were placed in the center chamber of a transparent 3-chamber arena and allowed 3 min to habituate to the space. The mouse was then placed into the closed center chamber and the trial was started by removing the doors and allowing 15 min to explore the 'social' or 'empty' chamber (Part I). The social chamber had a sex-matched, naïve mouse (4–5 weeks of age) in a cylindrical interaction cage, while the empty chamber had only the interaction cage. The chamber time and interaction time was recorded, with interaction time being defined as experimental mouse nose point being within interaction zone (5 cm from interaction cage). Immediately after Part I, a novel mouse, also sex-matched and naïve, was introduced in the empty cage and mice were allowed 15 min to explore 'Novel' or 'Familiar' chamber (Part II). The familiar mouse is the social mouse from Part I. The locations of the novel and familiar mice were switched to reduce side preference for this interaction partner.

## Barnes maze

The protocol was adapted from *Pitts, 2018*. The Barnes maze apparatus is a 122 cm-diameter, white-acrylic circle with 40 evenly spaced, 5 cm holes drilled 2.5 cm from the edge of the circle. This experiment was done in light room conditions to increase the motivation to escape. The volume of the white noise was also increased (90 dB). Four visual cues were placed on each side of the platform, which consisted of brightly colored shapes (purple diamond, yellow star, green cube, red triangle).

The mouse was first habituated to the escape tunnel, a small box filled with Alphadry bedding, in its home cage for 1 min. It was then habituated to the experimental conditions. The mouse was placed in the center of the apparatus and allowed to explore until it entered the escape tunnel or 5 min elapsed. If the mouse did not organically reach the escape tunnel within the time limit, it was led to the exit. There was at least 1 hr between habituation and the onset of acquisition training. The escape tunnel was moved between habituation and training trials.

There were four training trials per day for four consecutive days. There was an inter-trial interval of 30 min. During training trials, the position of the escape tunnel remained at a fixed location relative to the spatial cues. The mouse was placed in the center of the platform and tracked until it fully entered the escape tunnel or 5 min elapsed. If the mouse failed to reach the escape tunnel in the allotted time, it was led to the escape by the researcher. Following each trial, the platform was cleaned with 70% alcohol and the bedding inside the escape chamber was replaced.

For each trial, several variables were tracked to assess performance. These include distance traveled, latency to exit, incorrect hole visits (primary errors), and incorrect hole revisits (secondary errors). Errors are defined as a nose point entering a hole that does not contain the exit. Finally, based on these errors, the search strategy was characterized. Mice searched the maze serially, spatially, or randomly. Serial searching mice spent most of the time on the periphery performing a systematic search of the holes in a clockwise or counterclockwise manner, with 2 or less direction changes. Spatial search strategy was defined as <10 primary errors and a significantly reduced path and latency to exit. These mice use the surroundings to determine the shortest path to the exit and often are within 1–2 holes of the destination. All other results were defined as random search strategy. Random search strategy was usually characterized by multiple direction changes and skipping between holes, with many primary and secondary errors.

24 hr after the last training trial, mice were tested with a probe trial. The escape tunnel was removed before the mouse was placed on the platform for 5 min. Total time spent in each quartile was recorded, along with visits to former correct and incorrect escapes.

## Light/dark box

Mice were placed in the light side of a light/dark box apparatus and movement within the apparatus was recorded from above for 10 min. The 'light' side of the box had translucent walls and no ceiling, illuminated by the overhead room light. The 'dark' side had opaque walls and a visible-light-filtering ceiling, allowing the mouse to experience a dark environment but the infrared camera to continue recording. We recorded the latency to enter the dark, transitions between zones, and time spent in zones.

## Elevated plus maze

The elevated plus maze apparatus is a white '+'-shaped platform elevated 50 cm above the floor with oppositely positioned 'open' arms and 'closed' arms. The closed arms are enclosed by 30 cm high opaque walls, while the open arms have no railing. Each mouse was tested for 5 min after being placed onto the center platform facing an open arm. The number of open and closed arm entries and the time spent on the various sections of the EPM was recorded. The amount of exploration into the open arm is calculated to measure anxiety-like behavior.

## Stereotaxic surgery

Mice underwent stereotaxic surgery as described in *Ehinger et al., 2021*. Specifically, mice were anesthetized by vaporized isoflurane and were then head-fixed in a stereotaxic frame (David Kopf Instruments). The mice were between 5 and 6 weeks old. The experimental virus, or relevant control, was infused into the dorsal hippocampus (anteroposterior (AP) –2.3, mediolateral (ML)±1.7, dorsoventral (DV) –1.7 mm measured from bregma) using stainless steel injectors (33 gauge; Small Parts Incorporated) connected to Hamilton syringes (10 µl, 1701). Animals received 1 µl of virus bilaterally at a rate of 0.1 µl/min controlled by an automatic pump (Harvard Apparatus). After the infusion was complete, the injectors remained at the site for 10 min to allow diffusion of the virus. Mice were allowed to recover in their home cages for at least 3 weeks before testing to allow for maximal viral expression.

## Confirmation of viral expression

At the end of the experimental timeline, animals were euthanized via cervical dislocation and the brains were removed. The brains were placed on ice and dissected into 1 mm coronal sections. The fluorescent protein expressed by the virus (either GFP or mCherry) was visualized using an EVOS FL tabletop fluorescent microscope (Thermo Fisher Scientific). Images were taken for future reference. Animals that failed to exhibit fluorescence associated with viral overexpression were excluded from the study (4 mice).

## Statistical analysis

### Biochemical analysis

Parametric tests were performed on data deemed by the D'Agostino-Pearson omnibus (K2) test to be derived from a normally distributed population. Data were analyzed using a two-tailed t-test with Welch's correction for normal populations. Mann-Whitney tests were performed on data derived from non-normal populations. The results were determined to be statistically significant if the p-value was less than 0.05.

### Electrophysiology analysis

Parametric tests were performed on data deemed to be derived from a normally distributed population. Unpaired t-tests were performed on the average measurement over the experimental period. When multiple intensities were used, a two-way repeated measures ANOVA was performed. *Post hoc* pairwise comparisons (Tukey's) were performed after a significant result in the ANOVA. The results were determined to be statistically significant if the p-value was less than 0.05.

### Behavioral analysis

Parametric tests were performed on data deemed to be derived from a normally distributed population. Behavioral experiments were first assessed using a three-way ANOVA with primary variables being genotype, sex, and experimental variable where appropriate. Where there was no significant difference between sexes, the data was consolidated by genotype. A two-way ANOVA was then performed on the consolidated data (genotype × experimental variable). *Post hoc* Šidák's multiple comparison's test was performed to measure experimental differences directly. When there was only one experimental variable, an unpaired t-test was performed. If the variance was unequal, Welch's correction was applied. The results were determined to be statistically significant if the p-value was less than 0.05.

# Acknowledgements

We thank Carolyn Ferguson for expert technical assistance. Supported by NIH grants AA027474, AA027682 (DR) and AA020889 (GH). ZWH was partially funded by GM007175. None of the authors have a conflict of interest.

## Additional information

### Funding

| Funder | Grant reference number | Author |
|---|---|---|
| National Institute on Alcohol Abuse and Alcoholism | AA020889 | Gregg Homanics |
| National Institute on Alcohol Abuse and Alcoholism | AA027474 | Dorit Ron |
| National Institute on Alcohol Abuse and Alcoholism | AA027682 | Dorit Ron |
| National Institute of General Medical Sciences | GM007175 | Zachary W Hoisington |

The funders had no role in study design, data collection and interpretation, or the decision to submit the work for publication.

### Author contributions

Zachary W Hoisington, Data curation, Formal analysis, Investigation, Visualization, Methodology, Writing – original draft, Writing – review and editing; Himanshu Gangal, Formal analysis, Investigation; Khanhky Phamluong, Data curation, Formal analysis; Chhavi Shukla, Alexandra Salvi, Data curation; Jeffrey J Moffat, Investigation; Gregg Homanics, Conceptualization, Methodology; Jun Wang, Conceptualization, Formal analysis, Supervision, Validation, Investigation, Methodology, Writing – review and editing; Yann Ehinger, Conceptualization, Supervision, Validation, Investigation, Methodology, Writing – original draft, Writing – review and editing; Dorit Ron, Conceptualization, Supervision, Funding acquisition, Validation, Investigation, Methodology, Writing – original draft, Project administration, Writing – review and editing

### Author ORCIDs

Alexandra Salvi https://orcid.org/0000-0002-0148-2406
Jun Wang https://orcid.org/0000-0002-0085-4722
Yann Ehinger https://orcid.org/0000-0001-9314-6575
Dorit Ron https://orcid.org/0000-0001-5161-967X

### Ethics

All animal procedures were approved by UCSF Institutional Animal Care and Use Committee (IACUC) (animal protocol AN206967) and were conducted in agreement with the Association for Assessment and Accreditation of Laboratory Animal Care (AAALAC). Mice were euthanized by carbon dioxide inhalation followed by cervical dislocation, or deep anesthetization with pentobarbital (150 mg/kg) followed by transcranial perfusion with 4% paraformaldehyde. These procedures are in accordance with the Panel on Euthanasia of the American Veterinary Medical Association guidelines and with the standard operating procedures of the UCSF IACUC.

Reviewer #1 (Public review): https://doi.org/10.7554/eLife.100653.3.sa1
Reviewer #2 (Public review): https://doi.org/10.7554/eLife.100653.3.sa2
Author response https://doi.org/10.7554/eLife.100653.3.sa3

# Additional files

**Supplementary files**
MDAR checklist

**Data availability**
The authors declare that all relevant data supporting the findings of this study are included in this published article, supplementary information files and source data files. Source data are provided in this paper.

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
