## [Editor Report · eLife Assessment]

This **important** study aims to understand the function of ProSAP-interacting protein 1 (Prosapip1) in the brain. Using a conditional Prosapip1 KO mouse (floxed prosapip1 crossed with Syn1-Cre line), the authors performed analysis including protein biochemistry, synaptic physiology, and behavioral learning. **Convincing** evidence from this study supports a role of Prosapip 1 in synaptic protein composition, synaptic NMDA responses, LTP, and spatial memory.

---

## [Referee Report · Reviewer #1 (Public review)]

Summary:

Summary of what author's were trying to achieve: In the manuscript by Hoisington et al., the authors utilized a novel conditional neuronal prosap2-interacting protein 1 (Prosapip1) knockout mouse to delineate the effects of both neuronal and dorsal hippocampal (dHP)-specific knockout of Prosapip1 impacts biochemical and electrophysiological neuroadaptations within the dHP that may mediate behaviors associated with this brain region.

Strengths:

(1) Methodological Strengths

a) The generation and use of a conditional neuronal knockout of Prosapip1 is a strength. These mice will be useful for anyone interested in studying or comparing and contrasting the effects of loss of Prosapip1 in different brain regions or in non-neuronal tissues.

b) The use of biochemical, electrophysiological, and behavioral approaches are a strength. By providing data across multiple domains, a picture begins to emerge about the mechanistic role for Prosapip1. While questions still remain, the use of the 3 domains is a strength.

c) The use of both global, constitutive neuronal loss of Prosapip1 and postnatal dHP-specific knockout of Prosapip1 help support and validate the behavioral conclusions.

(2) Strengths of the results

a) It is interesting that loss of Prosapip1 leads to specific alterations in the expression of GluN2B and PSD95 but not GluA1 or GluN2A in a post homogenization fraction that the author's term a "synaptic" fraction. Therefore, these results suggest protein-specific modulation of glutamatergic receptors within a "synaptic" fraction.

b) The electrophysiological data demonstrate an NMDAR-dependent alteration in measures of hippocampal synaptic plasticity, including long-term potentiation (LTP) and NMDAR input/output. These data correspond with the biochemical data demonstrating a biochemical effect on GluN2B localization. Therefore, the conclusion that loss of Prosapip1 influences NMDAR function is well supported.

c) The behavioral data suggest deficits in memory in particular novel object recognition and spatial memory, in the Prosapip1 knockout mice. These data are strongly bolstered by both the pan neuronal knockout and the dHP Cre transduction.

The authors highlight potential future studies to further the understanding of Prosapip1.

---

## [Referee Report · Reviewer #2 (Public review)]

The authors provide valuable findings characterizing a Prosapip1 conditional knockout mouse and the effects of knockout on hippocampal excitatory transmission, NMDAR transmission, and several learning behaviors. Furthermore, the authors selectively and conditionally knockout Prosapip1 in the dorsal hippocampus and show that it is required for the same spatial learning and memory assessed in the conditional knockout mice. The study uncovers how Prosapip1 is involved PSD organization and is a functional and critical player in dorsal Hippocampal LTP via its interaction with GluN2B subunits. The study is well controlled, detailed, and data in the paper match the conclusions.

Comments on revisions:

The authors have addressed all concerns.

---

## [Author Response]

The following is the authors’ response to the original reviews.

**Reviewer #1 (Recommendations for the authors):**
The biochemical fractionation and use of the term "synaptic" were my biggest issues. I would recommend using a more targeted approach to measure the PSD or compare and contrast synaptic from extrasynaptic. For instance, PMID 16797717 does a PSD purification, whereas other papers have fractionated extrasynaptic from synaptic. Moreover, a PSD95 immunoprecipitation may be of interest as one question that could arise is since you see decreases in PSD95 GluN2B, but not 2A or GluA1, could the association of PSD95 with the different proteins be altered? To evaluate this, proteomics or some other unbiased methodology could enhance an understanding of the full panoply of changes induced by Prosapip1 within the dHP.

The reviewer makes value points; however, this is a large endeavor, which we will address in future experiments.

There seems to be a missed opportunity to really determine how Prosapip1 is influencing protein expression and/or phosphorylation at the PSD.

There is no indication that Prosapip1 is linked to transcription or translation machinery; therefore, we don’t see the value of examining protein expression in this context. Phosphorylation is a broad term, and although this can be answered through phosphoproteomics, this is outside the scope of this study.

At the very least, additional discussion within this realm would help the reader contextualize the biochemical data.

Further studies are needed to determine the mechanism by which Prosapip1 controls the localization of PSD95, GlunN2B, and potentially others. It is plausible that posttranslational modifications are responsible for Prosapip1 function. For example, the Prosapip1 sequence contains a potential glycosylation site (Ser622), and several potential phosphorylation sites (https://glygen.org/protein/O60299#Glycosylation, https://www.phosphosite.org/proteinAction.action?id=18395&showAllSites=true#appletMsg). These posttranslational modifications can contribute to the stabilization of the synaptic localization of GluN2B and PSD95.

We added to the discussion the paragraph above as well as the caveat that proteomic studies are needed for a comprehensive study of the role of Prosapip1 in the PSD.

Weaknesses:(1) Methodological Weaknessesa. The synapsin-Cre mice may more broadly express Cre-recombinase than just in neuronal tissues. Specifically, according to Jackson Laboratories, there is a concern with these mice expressing Cre-recombinase germline. As the human protein atlas suggests that Prosapip1 protein is expressed extraneuronally, validation of neuron or at least brain-specific knockout would be helpful in interpreting the data. Having said that, the data demonstrating that the brain region-specific knockout has similar behavioral impacts helps alleviate this concern somewhat; however, there are no biochemical or electrophysiological readouts from these animals, and therefore an alternative mechanism in this adult knockout cannot be excluded.

This is a valuable insight from the reviewer, especially considering the information from Jackson Laboratories. As mentioned in the paper, we exclusively used female Syn1-Cre carrying breeders to avoid germline recombination. Furthermore, we consistently assessed the prevalence of the Prosapip1 flox sites alongside the presence of Syn1-Cre with our regular litter genotyping, confirming the presence of Prosapip1. Additionally, Prosapip1 protein expression was directly examined in rats in Wendholdt et al., 2006, where this group reported that Prosapip1 is a brain-specific protein, minimizing the potential consequences of a peripheral loss of Prosapip1. In addition, to confirm that Prosapip1 is a brain-specific protein in mice, we performed a western blot analysis on the dorsal hippocampus, liver, and kidney of a C57BL/6 mouse (Author response image 1), and found that Prosapip1 protein is not found in these peripheral organs, aligning with the findings in rats reported by Wendholdt et al.

b. The use of the word synaptic and the crude fractionation make some of the data difficult to interpret/contextualize. It is unclear how a single centrifugation that eliminates the staining of a nuclear protein can be considered a "synaptic" fraction. This is highlighted by the presence of GAPDH in this fraction which is a cytosolically-enriched protein. While GAPDH may be associated with some membranes it is not a synaptic protein. There is no quantification of GAPDH against total protein to validate that it is not enriched in this fraction over control. Moreover, it should not be used as a loading control in the synaptic fraction. There are multiple different ways to enrich membranes, extrasynaptic fractions, and PSDs and a better discussion on the caveats of the biochemical fractionation is a minimum to help contextualize the changes in PSD95 and GluN2B.

We apologize for the confusion. As we described in the methods section, the crude synaptosome was isolated by several centrifugations as depicted in the figure which we are now including in the manuscript. As shown in Extended Figure 2, the P2 fraction does contain PSD-95 and synapsin, as well as GluN2B, GluN2A, and GluA1; however, it does not contain the transcription factor CREB, indicating the isolation of the crude synaptosomal fraction. As shown in the figure, a small amount of GAPDH is present in the crude synaptosomal fraction. The presence of GAPDH in the crude synaptosomal fraction has been previously reported in (Atsushi et al., 2003; Lee et al. 2016; Wang et al. 2012). As we have added to the discussion, there remains a caveat that we cannot differentiate the pre- and post-synaptic fraction, and as a result we do not know if Prosapip1 plays a role in the assembly of axonal proteins.

c. Also, the word synaptosomal on page 7 is not correct. One issue is this is more than synaptosomes and another issue is synaptosomes are exclusively presynaptic terminals. The correct term to use is synaptoneurosome, which includes both pre and postsynaptic components. Moreover, as stated above, this may contain these components but is most likely not a pure or even enriched fraction.

Since we cannot exclude the possibility that Prosapip1 is also expressed in glia, we do not believe that the term synaptoneurosome is accurate.

d. The age at which the mice underwent injection of the Cre virus was not mentioned.

We apologize for the oversight. As now noted in the methods, the mice used for experiments underwent surgery to infect neurons with the AAV-GFP or AAV-Cre viruses between 5 and 6 weeks of age to ensure full viral expression by the experimental window beginning at 8 weeks old.

(2) Weaknesses of Resultsa. There were no measures of GluN1 or GluA2 in the biochemical assays. As GluN1 is the obligate subunit, how it is impacted by the loss of Prosapip1 may help contextualize the fact that GluN2B, but not GluN2A, is altered. Moreover, as GluA2 has different calcium permeance, alterations in it may be informative.

Since we detect NMDAR current, which requires the obligatory subunit GluN1 and at least one GluN2 subunit (GluN2A, GluN2B, GluN2C, GluN2D), we did not see the rationale behind examining the level of GluN1 in the Prosapip1 knockout mice.

b. While there was no difference in GluA1 expression in the "synaptic" fraction, it does not mean that AMPAR function is not impacted by the loss of Prosapip1. This is particularly important as Prosapip1 may interact with kinases or phosphatases or their targeting proteins. Therefore, measuring AMPAR function electrophysiologically or synaptic protein phosphorylation would be informative.

We agree with the reviewer that the loss of Prosapip1 could potentially impact AMPAR function. To address this, we measured spontaneous excitatory postsynaptic currents (sEPSCs) in hippocampal pyramidal neurons from both Prosapip1(flx/flx);Syn1-Cre(-) and Prosapip1(flx/flx);Syn1-Cre(+) mice. Given that neurons were voltage-clamped at -70 mV and extracellular Mg^2+^ was maintained at 1.3 mM, the sEPSCs we recorded were primarily mediated by AMPARs.

We found no significant differences in either the frequency or amplitude of these AMPA-mediated sEPSCs between Prosapip1(flx/flx);Syn1-Cre(-) and Prosapip1(flx/flx);Syn1-Cre(+) mice, suggesting that AMPAR function in hippocampal pyramidal neurons is not noticeably affected by the loss of Prosapip1 (see Author response image 2 below).

**Author response image 2. sa3fig2:** 

c. There is a lack of mechanistic data on what specifically and how GluN2B and PSD95 expression is altered. This is due to some of the challenges with interpreting the biochemical fractionation and a lack of results regarding changes in protein posttranslational modifications.

See response above.

d. The loss of social novelty measures in both the global and dHP-specific Prosapip1 knockout mice were not very robust. As they were consistently lost in both approaches and as there were other consistent memory deficits, this does not impact the conclusions, but may be important to temper discussion to match these smaller deficits within this domain.

There is a clear difference between the Prosapip1(flx/flx);Syn1-Cre(-) and Prosapip1(flx/flx);Syn1-Cre(+) mice as well as the AAV-GFP and AAV-Cre mice in the loss of social novelty metric. We have emphasized that the Prosapip1(flx/flx);Syn1-Cre(+) mice and AAV-Cre mice do not recognize social novelty, which is supported by the statistics.

4E: Two-way ANOVA: Effect of Social Novelty F_(1,20)_ = 17.60, p = 0.0002; *Post hoc Familiar vs. Novel (Cre(-)) p = 0.0008, Familiar vs. Novel (Cre(+)) p = 0.1451.*

5I: Two-way ANOVA: Effect of Social Novelty F_(1,31)_ = 9.777, p = 0.0038; *Post hoc Familiar vs. Novel (AAV-GFP) p = 0.0303, Familiar vs. Novel (AAV-Cre) p = 0.1319.*

e. Alterations in presynaptic paired-pulse ratio measures are intriguing and may point to a role for Prosapip1 in synapse development, as discussed in the manuscript. It would be interesting to delineate if these PPR changes also occur in the adult knockout to help detail the specific Prosapip1-induced neuroadaptations that link to the alterations in novelty-induced behaviors.

This interesting question will be addressed in future studies.

**Reviewer #2 (Recommendations for the authors):**
(1) The test statistics are required for each experiment for completeness. Currently, only p-values, tests used, and N are included.

The entirety of the statistical information can be found in TYable 1, including test statistics and degrees of freedom (see Column 7, ‘Result’).

(2) The authors claim that the function of Prosapip1 is not known in vivo, yet detail a study in the NAc where they investigated its function in vivo. The wording or discussion around what is and is not known should be altered to reflect this.

The reviewer is correct to point to our previous manuscript (Laguesse et al. Neuron. 2017.) in which we found that Prosapip1 is important in mechanisms underlying alcohol-associated molecular, cellular and behavioral adaptations. However, these findings are specific to alcohol-related paradigms. Since the normal physiological role of Prosapip1 has never been delineated, this study was aimed to start addressing this gap in knowledge.

References

Wang, M., Li, S., Zhang, H. et al. Direct interaction between GluR2 and GAPDH regulates AMPAR-mediated excitotoxicity. Mol Brain 5, 13 (2012). https://doi.org/10.1186/1756-6606-5-13

Atsushi Ikemoto, David G. Bole, Tetsufumi Ueda, Glycolysis and Glutamate Accumulation into Synaptic Vesicles: Role of Glyceraldehyde Phosphate Dehydrogenase and 3-Phosphoglycerate Kinase, Journal of Biological Chemistry, 8, 278 (2003). https://doi.org/10.1074/jbc.M211617200.

Lee, F., Su, P., Xie, YF. et al. Disrupting GluA2-GAPDH Interaction Affects Axon and Dendrite Development. Sci Rep 6, 30458 (2016). https://doi.org/10.1038/srep30458